# MLLM as Retriever: Interactively Learning Multimodal Retrieval for Embodied Agents

**Junpeng Yue**[1]*, **Xinrun Xu**[2], **Börje F. Karlsson**[3], **and Zongqing Lu**[1]†

[1]School of Computer Science, Peking University
[2]Institute of Software, Chinese Academy of Sciences
[3]Beijing Academy of Artificial Intelligence

## Abstract

MLLM agents demonstrate potential for complex embodied tasks by retrieving multimodal task-relevant trajectory data. However, current retrieval methods primarily focus on surface-level similarities of textual or visual cues in trajectories, neglecting their effectiveness for the specific task at hand. To address this issue, we propose a novel method, **MLLM As ReTriever (MART)**, which enhances the performance of embodied agents by utilizing interaction data to fine-tune an MLLM retriever based on preference learning, such that the retriever fully considers the effectiveness of trajectories and prioritize them for unseen tasks. We also introduce Trajectory Abstraction, a mechanism that leverages MLLMs' summarization capabilities to represent trajectories with fewer tokens while preserving key information, enabling agents to better comprehend milestones in the trajectory. Experimental results across various environments demonstrate our method significantly improves task success rates in unseen scenes compared to baseline methods. This work presents a new paradigm for multimodal retrieval in embodied agents, by fine-tuning a general-purpose MLLM as the retriever to assess trajectory effectiveness. All the code for benchmark tasks, simulator modifications and the MLLM retriever is available at https://github.com/PKU-RL/MART.

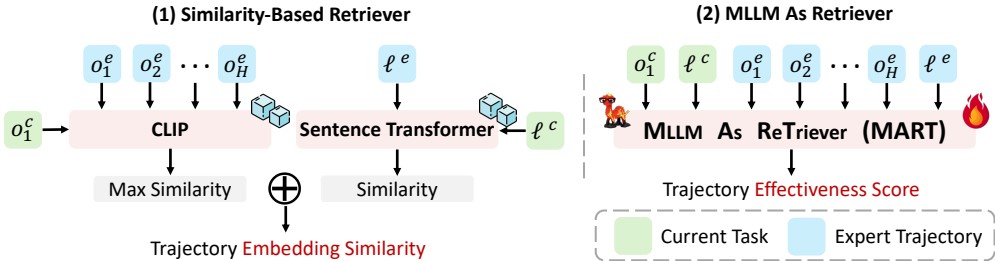

Figure 1: Similarity-Based Retriever vs. **MART**. Traditional multimodal retrieval methods (1) depend on calculating weighted sums of image and text embedding similarities, while our approach (2) introduces interactive learning to assess the relevance between the current and expert trajectories.

## 1 Introduction

Embodied agents interacting with complex environments require understanding both the current context and task-specific domain knowledge to perform effectively (Wang et al., 2023c; Lifshitz et al., 2023). Recently, Multimodal Large Language Models (MLLMs), which are capable of processing both textual and visual data, have shown promise in various embodied tasks – *e.g.,* table manipulation (Handa et al., 2023; Chen et al., 2022a), robot navigation (Zhang et al., 2024b; Shah et al., 2022), and 3D games (Wang et al., 2024a; Tan et al., 2024; Jiang et al., 2024). However, such

---

*Junpeng Yue and Xinrun Xu work as interns at BAAI.
†Correspondence to Zongqing Lu <zongqing.lu@pku.edu.cn>.

models typically lack effective grounding in the embodied environments in which agents operate, greatly limiting their performance in embodied tasks (Long et al., 2024; Wang et al., 2024c).

To mitigate this limitation, providing additional task-relevant grounding information is essential to better leverage the general capabilities of MLLMs. Trajectory data, consisting of sequences of actions and observations, can be easily available and provide valuable insights into task execution (Zheng et al., 2024; Zhao et al., 2024), therefore serving as a good information source for grounding. By using trajectory data in prompting an MLLM, the embodied agent can readily leverage previous experiences to better guide agents through similar tasks in new situations or environments (Zhang et al., 2024a; Lee et al., 2024). However, retrieving the most effective trajectories — those that can significantly enhance task performance — remains a challenge, particularly when multiple trajectories appear similar in both textual and visual modalities (Jeurissen et al., 2024).

Existing retrieval methods mainly focus on surface-level textual or visual similarities of trajectories, often neglecting key aspects critical for task effectiveness, *e.g.,* a trajectory with a similar task instruction but in a different scene, or one in the same scene but with a different layout. In such cases, these trajectories fail to provide useful information for the current task and can mislead the agent. As shown in Figure 2, relying solely on similarity is not effective in retrieving useful trajectories, as similarity does not directly correlate with success rate. To better support agents in embodied tasks, a trajectory retriever model needs to consider the effectiveness of trajectories for a given task.

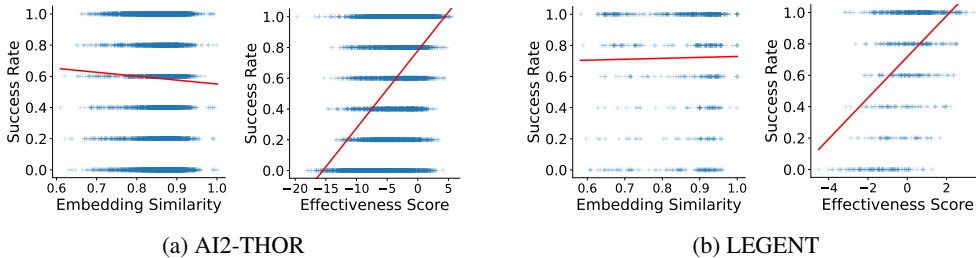

(a) AI2-THOR                                           (b) LEGENT

Figure 2: Scatter plots illustrating the relationship between success rate and embedding similarity (left) or effectiveness score (right) in two environments. The red line indicates a linear fit to the data.

To achieve a better retriever, we propose a new paradigm that integrates interactive learning with the retriever. Firstly, we consider expert trajectories of training scenarios as prompt for an MLLM agent, and let the agent interact with the environment to collect different success rates for different such reference trajectories. This interactive feedback data is then organized into preference pairs, which are used to fine-tune an MLLM – LLaVA (Liu et al., 2023) in our case – with a Bradley-Terry model (Bradley & Terry, 1952), such that the fine-tuned retriever model is capable of prioritizing more effective trajectories for unseen tasks. Combining this functionality with the inherent general capabilities of MLLM allows embodied agents to operate more effectively in unseen environments by leveraging their most useful past experiences.

We also introduce a new Trajectory Abstraction mechanism, which uses MLLMs' summarization capabilities to represent trajectories in a reduced number of tokens, while preserving key information and enabling agents to better understand such information in the trajectory (*e.g.,* key relevant overarching actions). This mechanism is especially important in long-horizon tasks, both reducing the required context window length and removing distracting information from trajectory samples.

Combining the aforementioned components, we present our approach – **MART** (**M**LLM **A**s **ReT**riever) – which adapts embodied agents in unseen scenarios by fine-tuning MLLM through preference data. To assess the benefits of our method, we conduct empirical experiments across diverse environments. The experimental results show that **MART** achieves significantly higher task success rates compared to baselines, demonstrating its effectiveness. With this approach, we present a new paradigm for multimodal retrieval in embodied agents, fine-tuning a general-purpose MLLM as a retriever capable of considering trajectory effectiveness. Our contributions can be summarized as follows:

- To the best of our knowledge, **MART** is the first approach that integrates interactive learning with a retriever and uses interactive feedback to fine-tune an MLLM retriever in eval-

uating trajectory effectiveness, combining its inherent general capabilities with the ability to assess the task-guiding effectiveness of trajectories.

- We introduce Trajectory Abstraction, a new mechanism that utilizes MLLM capabilities to significantly condense trajectories. This method reduces the token number while retaining essential information, allowing agents to effectively use this condensed knowledge in novel situations and provide guidance for long-horizon tasks.

- The effectiveness of **MART** is empirically validated through comprehensive experiments in various environments, demonstrating significant performance improvements on unseen tasks. **MART** consistently surpasses baselines by over 10% across different environments.

## 2 RELATED WORK

### 2.1 EMBODIED AGENTS BASED ON LARGE MODELS

Recently, there have been several attempts to utilize the general-purpose capabilities of large models for complex embodied tasks. These efforts can be broadly categorized into two types: VLA models and LLM/MLLM-based agents. **1) VLA models**, including PaLM-E (Driess et al., 2023), RT-2 (Brohan et al., 2023), Gato (Reed et al., 2022), VIMA (Jiang et al., 2022), and MOO (Stone et al., 2023), rely on trajectory data to train a Transformer-based VLM for action planning, without explicitly constructing a memory. However, their generalization capabilities are limited due to the inherent issue of catastrophic forgetting in neural networks. **2) LLM/MLLM-based agents**, like Voyager (Wang et al., 2024a) and DEPS (Wang et al., 2023b) for Minecraft, Cradle (Tan et al., 2024) for RDR2, LLM-Planner (Song et al., 2023) for ALFRed, and Code-as-Policies (Liang et al., 2023) for real-world embodied control, do not involve directly training new models. Instead, they leverage the general-purpose capabilities of LLM/MLLM primarily through prompt engineering. Most of these agents build and maintain comprehensive memory systems to assist in task completion. However, memory retrieval mostly focuses on surface-level similarity, overlooking actual effectiveness in completing complex tasks.

### 2.2 MEMORY RETRIEVAL IN AGENTS

Agents can continuously learn and improve by recalling task-related experiences (Zhang et al., 2024c; Xi et al., 2023). During interactions with the environment, two main types of information are stored in memory. **1) Semantic information:** Early LLM agents faced limitations due to input token constraints, leading to reliance on short-term memory and greedy strategies (Chen et al., 2024; Zhang et al., 2023; Abdelnabi & et al, 2023; Wang et al., 2023a). However, summarizing memory over short periods risked information loss (Light et al., 2023; Kaiya & et al, 2023). While storing comprehensive memory (semantic, episodic, procedural) can provide great value, its effective utilization for decision-making remains challenging (Sumers et al., 2024). More recent approaches try to prioritize relevant memories based on embedding similarity or LLM-based relevance identification (*e.g.,* (Park & et al., 2023), (Hong & et al., 2023), (Lin et al., 2023a), (Wang et al., 2024b), and (Xu et al., 2023)). **2) Image information:** Recent advances in MLLM agents have greatly enhanced their grounding abilities by allowing image memory retrieval during actions. As illustrated in Figure 1, Similarity-Based Retrievers for agents, *e.g.,* (Zhou et al., 2024; Wang et al., 2023c; Li et al., 2024), leverage image embedding. while Groot (Cai et al., 2023) encodes visual and temporal information from video frames for guiding actions. But neither considers direct task effectiveness.

## 3 INTERACTIVELY LEARNING MULTIMODAL RETRIEVAL

### 3.1 PROBLEM FORMULATION

In this study, we investigate interactions of a retrieval-augmented MLLM agent with an environment to complete embodied tasks drawn from a specific distribution. Figure 3 provides an overview of our **MLLM As ReTriever** approach – **MART**. The agent is assigned a task instruction $\ell^c$ sampled from the task instruction distribution $p(\ell)$, and operates over a finite horizon $H$. At each timestep $t \in 1, 2, ..., H$, the agent selects an action $a_t$ from the action space $A$ based on the current observation $o_t^c$ from the observation space $O$ and a reference trajectory $\tau^e$ retrieved from the expert trajectory

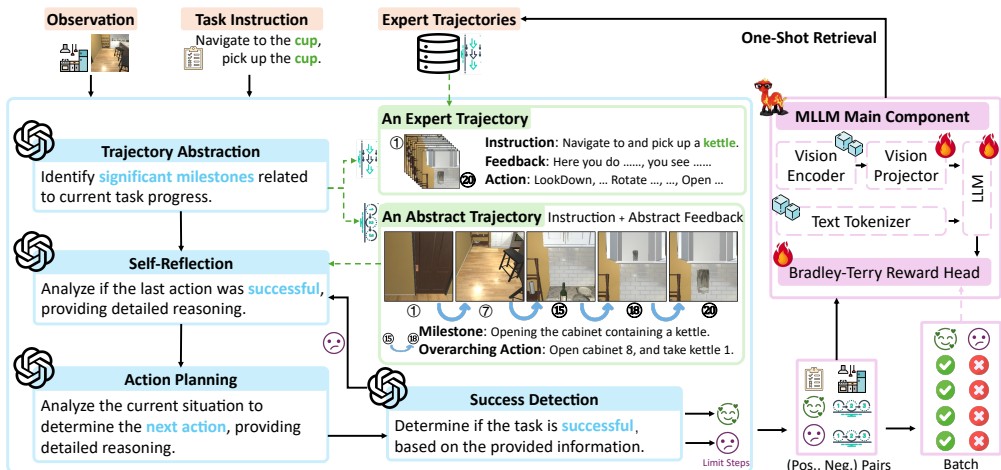

Figure 3: Overview of **MART**. Our approach interactively learns a multimodal retriever to score expert trajectories and retrieve most effective trajectory to guide an agent in novel situations. By considering trajectories with higher success rates as positive samples and those with lower success rates as negative trajectories, we obtain the preference pairs, which are used to fine-tune an MLLM retriever to score trajectory effectiveness for a specific task.

memory $\mathcal{M}$. The memory contains a set of multimodal expert trajectories $\mathcal{M} = \{\tau_1^e, \tau_2^e, ..., \tau_n^e\}$, where each trajectory $\tau_i^e$ includes task instructions $\ell_i^e$ sampled from the same task instruction distribution $p(\ell)$. The trajectory $\tau_i^e$ also contains observation sequences $\vec{o_i^e} = \{o_{i_1}^e, o_{i_2}^e, ..., o_{i_H}^e\}$, and action sequences $\vec{a_i^e} = \{a_{i_1}^e, a_{i_2}^e, ..., a_{i_H}^e\}$.

The agent follows a frozen policy $\pi(a|\ell^c, \tau^e, o^c)$, implemented as a Multimodal Large Language Model (MLLM), and the reference trajectory $\tau$ plays a significant role in grounding the agent within the embodied environment, supporting task accomplishment. This reference trajectory is retrieved through an MLLM retriever $q_\theta$.

We fine-tune our MLLM retriever on training task distribution $p_{\text{train}}(\ell)$ and evaluate its performance on test task distribution $p_{\text{test}}(\ell)$, which has no overlap with training tasks. This retriever aims to identify and retrieve a trajectory $\tau^e$ from the expert memory pool that is most effective for the current task $\ell^c$, *i.e.,* which can help ground the MLLM agent with a specific embodied task and enable its effective completion. It is worth noting that we have different memories $\mathcal{M}^{\text{train}}$ and $\mathcal{M}^{\text{test}}$, which corresponds to training tasks and test tasks, respectively.

## 3.2 MEMORY

To enable trajectory retrieval for task execution and fine-tuning our MLLM retriever, we first construct memory databases containing expert trajectories from previous successful executions for tasks both from $p_{\text{train}}(\ell)$ and $p_{\text{test}}(\ell)$. For each trajectory, $\tau_i^e = \{\ell_i^e, \vec{o_i^e}, \vec{a_i^e}\}$, represents a task $\ell_i^e$ completed in $H_i$ steps and comprises the sequence of observations $\vec{o_i^e}$, and corresponding actions $\vec{a_i^e}$.

In multi-modal environments, such as AI2-THOR (Kolve et al., 2017) and LEGENT (Cheng et al., 2024), each timestep observation $o_i$ includes an egocentric image. Moreover, in the AI2-THOR environment, we assign numerical IDs (e.g. Cup 1) to all objects in the current visible field of view to identify target objects for interaction. These IDs appear in the environment feedback output in natural language, which is also part of the observation.

Expert trajectories are collected via a planner-based method (Hoffmann & Nebel, 2001). Storing these trajectories allows the agent to later leverage past experience when facing new task instances. Trajectory data is collected independently for the training and test sets. For each task, we initialized a task instance and used a planner-based method to collect expert trajectory data. It is worth mentioning that the initialization position and orientation in each task is randomly chosen.

Since each task is directly corresponds to one unique task in the task set, the size of our memory used for experiments is relatively small. For example, in the experiments performed in the LEGENT environment, the memory pool consisted of 40 trajectories for training, and distinct 32 trajectories during testing. More details are available in the experimental settings (Section 4.1).

## 3.3 MULTIMODAL RETRIEVER

The core of **MART** is the innovative use of interactive learning to train the trajectory retriever. For an embodied task, different trajectories stored in memory can be provided as references to the MLLM agent, leading to varying effects on the completion of the current task, depending on the degree of grounding with the environment they provide. Even if a trajectory has text instruction similar to the current task or an image sequence similar to the initial egocentric observation of the task, it does not guarantee that this trajectory can provide effective grounding. This is due to plain similarity alone not being able to reflect the effectiveness of the trajectory for the embodied task. For instance, a failed trajectory for a related task could have high textual and visual similarity to the target task.

In order to retrieve the trajectory that can provide the most benefit (*i.e.,* effective grounding) for the current task from the trajectory memory, we propose an interactive learning method for the MLLM retriever. Specifically, for each task in training set, we sample $K$ trajectories from the training memory $\mathcal{M}^{\text{train}}$, and feed them as prompt for MLLM agent to execute the embodied task respectively. After that, based on the induced success rates of task execution, we can get the effectiveness of each trajectory for the embodied task. We can then obtain a partial order list based on success rate comparisons, producing $\binom{K}{2}$ pairs through pairwise comparison, where the trajectory with a higher success rate is treated as the positive item, and the one with a lower success rate as the negative item. In this way, these preference pairs from interactive feedback are arranged as a positive-negative pair dataset $D$, which we use to fine-tune the MLLM according to the Bradley-Terry (Bradley & Terry, 1952) Reward Modeling loss to enhance its critiquing ability, as in Equation 1:

$$\mathcal{L}(\theta) = -\mathbb{E}_{(\ell_1^c, o_1^c, \mathcal{A}(\tau_w^e), \mathcal{A}(\tau_l^e)) \sim D} \left[ \log \left( \sigma \left( q_\theta(\ell^c, o_1^c, \mathcal{A}(\tau_w^e)) - q_\theta(\ell^c, o_1^c, \mathcal{A}(\tau_l^e)) \right) \right) \right]. \quad (1)$$

In particular, we use LLaVA-7B (Liu et al., 2023) as base MLLM and add a Bradley-Terry score head based on hidden states of base model output. The score head is a one-layer MLP that takes as input the last token in the hidden state and outputs a scalar score. The input to the MLLM retriever includes the trajectory abstraction result of expert trajectory $\tau_i^e$, $\mathcal{A}(\tau_i^e)$, current observation $o_1^c$, current task instruction $\ell^c$, and a prompt for it to judge the effectiveness of this trajectory for the current task and state. Upon inference, the MLLM outputs the score of the trajectory, which indicates its effectiveness for the current task. We select the trajectory with the highest score as the reference for the embodied agent to complete the current task.

## 3.4 TRAJECTORY ABSTRACTION

A complete multimodal trajectory often has dozens of timesteps, which correspond to dozens of observations, actions, and feedback, including redundant or irrelevant information. Furthermore, inputting all trajectory tokens, especially image tokens, to the MLLM retriever and agent will likely lead to confusing the models/agent or even exceeding their context windows.

We thus use another MLLM (in our experiments, GPT-4o) to automatically create an abstract trajectory in zero-shot manner. The initial input to the MLLM is the trajectory $\tau^e$ (consisting of task instruction $\ell^e$, observation sequence $\vec{o^e}$, and action sequence $\vec{a^e}$) as well as the current task instruction $\ell^c$, and we let the MLLM find whether each observation contained in the trajectory $\tau^e$ is helpful for the current task $\ell^c$. If it is considered to be useful, we will keep the observation into the resulting trajectory abstraction $\mathcal{A}(\tau^e)$.

To be more specific, we let the MLLM comprehend the tasks accomplished in the given trajectory $\tau^e$, and then identify important observations in the trajectory as milestones and preserve them into the resulting trajectory abstraction. These milestones are steps that are essential for accomplishing the trajectory task of $\ell^e$, such as steps where important decisions are made, goals are achieved, or notable changes in the environment or state occur. Also if the target object of current task instruction $\ell^c$ appears in the trajectory feedback, then the step where it appears is also considered to be a significant milestone as it is strongly related to current task $\ell^c$. The milestone output format consists

of: 1) a description of the milestone; 2) the corresponding image ({image x}); 3) the corresponding feedback ({feedback x}); and 4) the overarching actions taken between this milestone and the next one.

The summarized trajectory manages to remove redundant information without affecting relevance for the current task $\ell^c$, so that the agent can better receive grounding information contained in the trajectory. Quantitatively, for the tasks in the test set, the input trajectories of trajectory abstraction have an average of 11.51 steps, while the output milestones have an average count of 3.13.

## 4 EXPERIMENTS

### 4.1 EXPERIMENTAL SETUP

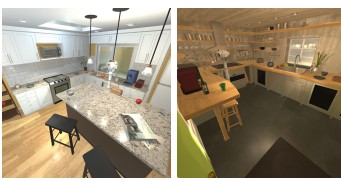

(a) AI2-THOR

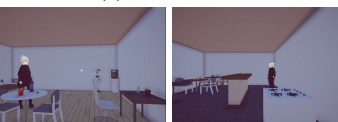

(b) LEGENT

Figure 4: Environment comparison.

Table 1: Environment complexity comparison. AI2-THOR contains more than four times the number of interactive objects per scene, and more complex object hierarchy, compared to LEGENT. E.g., AI2-THOR supports relationships such as "inside" (e.g., inside a microwave (Figure 10a), or inside an open container, like a sink (Figure 10c)).

|                   | AI2-THOR        | LEGENT |
|-------------------|-----------------|--------|
| Avg. Objects/Scene | 47.30          | 11.13  |
| Object Hierarchy   | Inside, On      | On     |
| Layout Complexity  | High            | Low    |
| Task decomposition | Yes             | No     |
| Observation        | Image, Feedback | Image  |

#### 4.1.1 ENVIRONMENTS

To validate the effectiveness of our method in various environments, we perform evaluations on multiple scenarios in two environments, AI2-THOR (Kolve et al., 2017) and LEGENT (Cheng et al., 2024). Unlike LLM agents, which use only text as input and simplify the action space by employing teleportation actions (*e.g.,* "move directly to target"), we believe that MLLM agents should undertake more challenging tasks as they have access to visual input and are no longer limited to a 'blind' mode of operation. Therefore, both target environments are multimodal, whose observations are egocentric images, and both make use of fine-grained control actions.

**AI2-THOR** simulates embodied household tasks supporting natural language instructions and egocentric visual observations, where agents must navigate and interact with various household items within realistic 3D environments, including kitchens, living rooms, and other indoor spaces. it allows agents to perform fine-grained navigation actions including 'move ahead', 'turn left/right x degrees', 'look up/down', and interactive atomic action including 'pick up object A', 'put object A on/in object B', 'open/close object A', 'toggle on/off object A'.

**LEGENT** is designed to imitate human activities and tasks in home environments, including cross-room navigation. The action space is similar to AI2-THOR's, including fine-grained movement actions. It also includes a 'speak' action, which sends a message to the user.

Table 1 and Figure 4 show a comparison between the two environments. Besides having many more objects in its scenes, AI2-THOR supports more complex object hierarchy relationships, such as "inside" (*e.g.,* "inside a cabinet", which require open/close interaction to complete a task, or "a cup inside a sink", *i.e.,* an open container), not supported in LEGENT [1].

#### 4.1.2 TASK SETTINGS

Notably, in all tasks, the initial position and orientation of the agent is chosen randomly. Each task is tested 5 times to reduce the impact of random errors.

---

[1] Although there are cabinets in LEGENT, they do not need to be opened/closed to complete tasks.

**AI2-THOR**. To better assess the fine-grained control ability of MLLM agents to complete real-world embodied tasks, we integrate characteristics of two AI2-THOR-based benchmarks – ALF-World (Shridhar et al., 2021) and ALFRed (Shridhar et al., 2020) – and built an environment setting that is more suitable for the MLLM agent. Since tasks are long-horizon, we follow the method in ALFRed and apply task decomposition to divide them into sub-tasks before execution. Each sub-task is then provided to the agent, and it determines sub-task success based on environmental feedback by itself (details in Appendix F.2 and F.3 ). Once a sub-task is successfully completed, the agent proceeds to the next sub-task. Task types include `pick_and_place`, `pick_clean_then_place`, `pick_cool_then_place`, and `pick_heat_then_place`. Completing these tasks requires dozens of steps of navigation, as well as interaction with objects. There are 45 tasks comprising a total of 260 sub-tasks in training set, and 28 tasks including 158 sub-tasks in testing set.

**LEGENT**. Tasks for the MLLM agent are categorized into two types: 'Come Here' and 'Where Is'. Each task is further divided into 'One-room' and 'Two-room' types, based on whether it requires traversing between rooms. In LEGENT, task decomposition is not performed as task instructions are simpler and do not contain combinations of sub-tasks; *i.e.,* the granularity of tasks is similar to that of sub-tasks in AI2-THOR. To train the retriever, we use 40 tasks (10 tasks for each task type) and we use 32 tasks, also covering all task types, as test set.

### 4.1.3 MEMORY CONSTRUCTION

Memory initialization follows the procedure described in Section 3.2; with randomized starting positions.

**AI2-THOR**. Once trajectories are collected, we decompose them into sub-task trajectories (following the task decomposition procedure in ALFRed) and treat each sub-task level trajectory as a expert trajectory into memory. Similar redundant trajectories are then filtered out, accounting for about one-third of total, resulting in a collection of 170 memory trajectories for the training memory and 118 trajectories for the testing memory.

**LEGENT**. As decomposition is not necessary due to the simpler tasks in this environment, the training memory is initialized with 40 trajectories, and the test memory with distinct 32 trajectories; one trajectory per task.

### 4.1.4 TASK EVALUATION

To evaluate the effectiveness of the retrieval-augmented embodied agents, we assess their performance using two metrics: Success Rate (**SR**) and Average Steps (**AS**).

Success Rate denotes the percentage of tasks attempts successfully completed by the agent. In AI2-THOR it indicates the percentage of completed full tasks, and we additionally use **SR-Sub** to represent the percentage of completed sub-tasks.

Average Steps represents the average number of steps the agent takes to complete a task. In AI2-THOR it indicates the number of steps to complete a full task, and **AS-Sub** represents the step average to complete a sub-task. Notably, for failed cases, their steps are counted as the step limit.

### 4.1.5 BASELINES

It is worth noting that **MART** is the first work to retrieve multimodal trajectories as references for embodied MLLM agents and let the agent directly output fine-grained control actions. We compare **MART** against three baseline methods to explore the performance of our approach:

**Plain-Agent (PA)** is an embodied MLLM agent without making use of reference trajectories, *i.e.,* without any memory. In our experiments, we use GPT-4o (2024-05-13 version).

**LLaVA-Plain (LP)** is a pre-trained LLaVA with no modified head and no finetuning. We use the probability of special token generation to represent the score. Its input is the same as **MART**, and it is prompted to output only Yes/No tokens, and the final score is calculated based on the probability of token generation (more details and limitations in Appendix E.3).

Table 2: Performance comparison of different methods in AI2-THOR.

|  | PA | LP | SL | RAP | MART |
|---|---|---|---|---|---|
| **SR ↑** | 0.18 | 0.26 | 0.24 | 0.22 | **0.40** |
| **SR-Sub ↑** | 0.63 | 0.69 | 0.68 | 0.67 | **0.75** |
| **AS ↓** | 159.66 | 144.18 | 147.48 | 147.03 | **123.19** |
| **AS-Sub ↓** | 44.65 | 39.88 | 40.79 | 40.67 | **34.07** |

Table 3: Performance comparison of different methods in LEGENT.

|  | PA | LP | SL | RAP | MART |
|---|---|---|---|---|---|
| **SR ↑** | 0.70 | 0.69 | 0.75 | 0.75 | **0.87** |
| **AS ↓** | 23.62 | 25.01 | 20.92 | 20.62 | **13.81** |

Table 4: Ablation studies of **MART** in the AI2-THOR and LEGENT environments.

| Environment | Metric | w/o Abstraction | Sim.+FTM | MART |
|---|---|---|---|---|
| **AI2-THOR** | **SR ↑** | 0.31 | 0.34 | **0.40** |
|  | **SR-Sub ↑** | 0.73 | 0.74 | **0.75** |
|  | **AS ↓** | 130.26 | 125.20 | **123.19** |
|  | **AS-Sub ↓** | 36.03 | 34.63 | **34.07** |
| **LEGENT** | **SR ↑** | 0.77 | 0.77 | **0.87** |
|  | **AS ↓** | 18.48 | 18.83 | **13.81** |

**Similarity+LLaVA (SL)** is a reasonable retrieval+ranking approach. Such approach is common in the text retrieval field – *e.g.,* Sun et al. (2023b;a); Dong et al. (2024) – and it can take into account both similarity and effectiveness. We use similarity to choose the top-K candidate trajectories, and then choose the most likely effective one using a plain LLaVA model (*i.e.,* same as **LP**).

**RAP** (Kagaya et al., 2024) performs retrieval based on plain similarity per modality and is the most similar setting in literature. It mainly targets text modality experiments in the ALFWorld (Shridhar et al., 2021) and WebShop (Yao et al., 2022) (treated as text-only) environments, and simple tasks in the multimodal environments Franka-Kitchen (Gupta et al., 2019) and Meta-World (Yu et al., 2019).

## 4.2 AI2-THOR

We firstly demonstrate the effectiveness of **MART** over test tasks in the AI2-THOR environment. The experimental results demonstrate the effectiveness of our approach compared to the baselines. As shown in Table 2, **MART** surpasses all baselines over 10% in Success Rate, and reaches best performance across all metrics.

## 4.3 LEGENT

We then conduct experiments in the LEGENT environment. This environment includes tasks involving crossing between rooms, thereby enriching the experimental space and demonstrating the effectiveness of our method. The experimental results, shown in Table 3, demonstrate **MART** greatly surpasses all baselines in all four task types.

## 4.4 ABLATIONS

In this section, we use a set of ablation studies to examine the contribution of key components in **MART**. More specifically, we aim to answer the following questions.

**Q1.** Does Trajectory Abstraction indeed improve embodied agent performance?

We compare the full **MART** approach and the ablation of removing its Trajectory Abstraction module (**w/o Abstraction**). As shown in Table 4, **MART** consistently reaches the best results across settings. Even if in AI2-THOR sub-tasks the average success rate is comparable, the improvement margin leads to a 9 percentage points improvement in full task success rate.

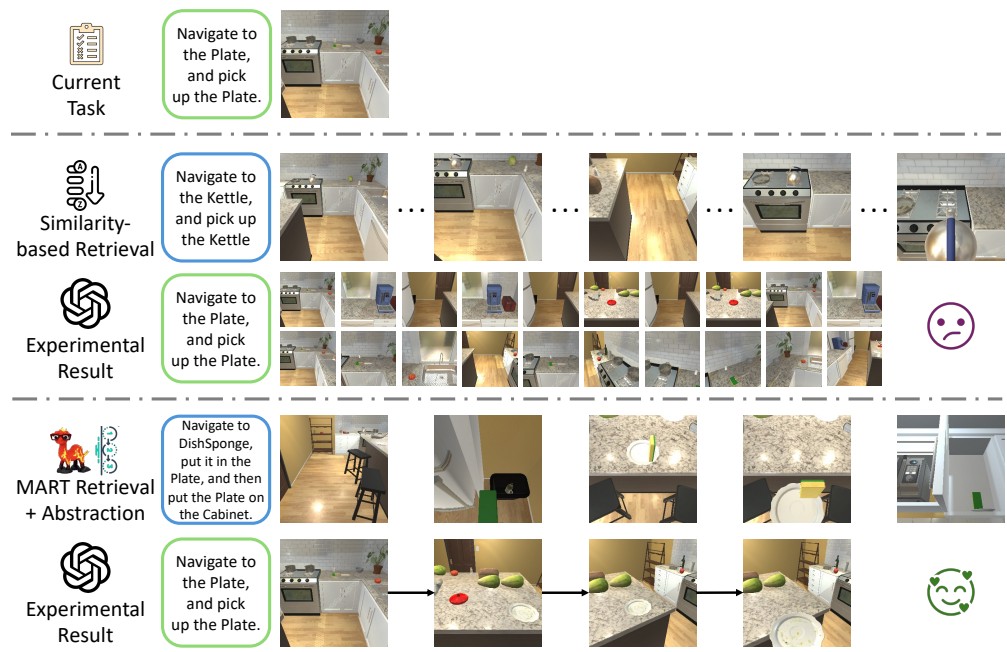

Figure 5: Comparison between similarity-based retriever and **MART**.

**Q2.** How does the **MART** approach compare against a typical retrieve and rank approach, even if it uses a fine-tuned ranking model for trajectory usefulness?

Table 4 shows decomposing the unified **MART** approach into separate retrieval and ranking steps (*i.e.*, similarity-based retrieval and **MART**'s MLLM as ranker – **Sim.+FTM**) decreases success rates across settings. It is also interesting to note that **Sim.+FTM** outperforms both **SL** and **RAP** (Tables 2 and 3), further illustrating **MART**'s trajectory utility scoring.

## 4.5 CASE STUDY

We present two case studies for more in-depth discussion of **MART**'s capabilities handling challenges of the MLLM agent setting. The first case (Figure 5) demonstrates the effective handling of a very long-horizon trajectory. Given a 73-step task trajectory – "navigate to DishSponge, put it in the Plate, and place the Plate on the Cabinet" – Trajectory Abstraction identifies 5 key milestones. **MART** achieves an 80% success rate with an average of 28 steps, while both the similarity-based method and the agent without memory reach only 40% success rate, averaging 69.6 and 74.6 steps, respectively. We provide more detailed and balanced case studies in Appendix D.

The second case (Figure 6) shows how **MART** extracts implicit rules for long sequence tasks. For the task "put the Potato into the microwave, heat it, and pick it up", Trajectory Abstraction analyzes the agent's actions (including exploration, attempts, and success) and generates an abstract set of inferred rules, pruning non-contributory and redundant actions, reducing 13 transitions to just 6. **MART** achieves an 80% success rate with an average of 30.2 steps, while other methods have a 0% success rate.

## 5 CONCLUSION

We propose **MART**, a new paradigm for trajectory retrieval incorporating interactive learning, to enhance embodied agents' performance by providing them with task-relevant trajectory data. Our approach utilizes interaction-based feedback to identify the most effective trajectories, and constructs preference pairs based on the comparisons between trajectories. An MLLM retriever is fine-tuned through these preference pairs, effectively prioritizing the trajectories that improve task performance. We also introduce Trajectory Abstraction in **MART**, a novel mechanism that leverages

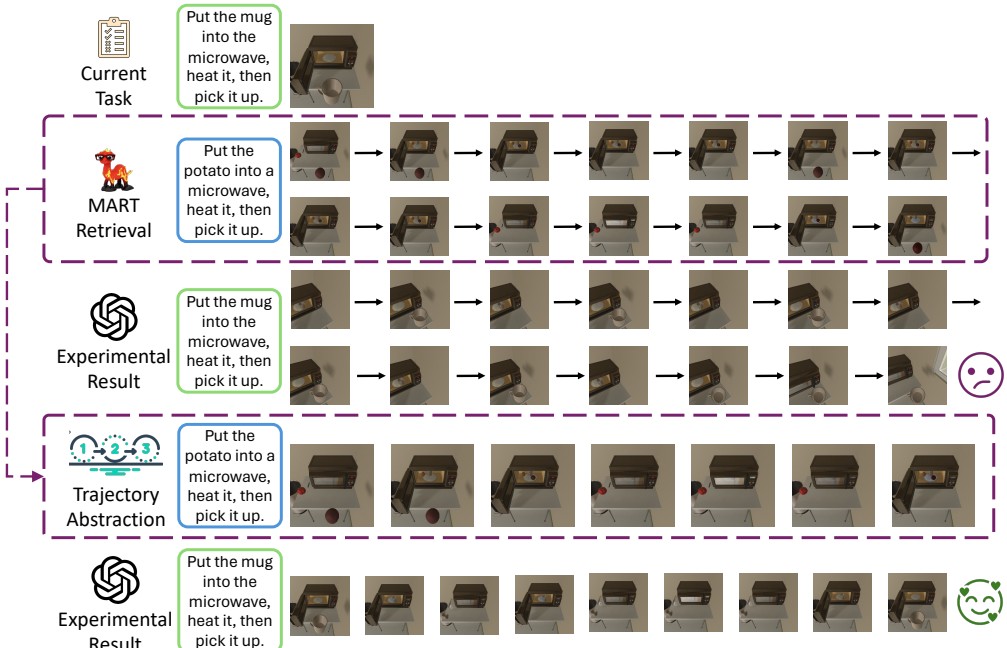

Figure 6: Showcase of the significance of the Trajectory Abstraction mechanism.

MLLMs' summarization capabilities, to abstract trajectories, *i.e.,* reduce the required number of tokens to represent them, while preserving key information and enabling agents to better understand relevant information. Experimental results in different environments demonstrate that our method significantly enhances task success rates in unseen tasks, compared to multiple baselines. This work helps bridge the gap between general-purpose MLLMs and the specific requirements of embodied tasks, offering a new paradigm for multimodal trajectory retrieval for embodied agents.

# 6 DISCUSSIONS AND FUTURE WORKS

Our study has a few limitations. First, the MLLM's restricted context window limits its ability to process multiple images simultaneously, restricting us to one-shot learning with single trajectory inputs. Future work will explore few-shot learning to combine skills from multiple trajectories for enhanced performance in complex tasks.

Second, the lack of a detailed ablation study will be addressed by evaluating each component's contribution, including self-reflection mechanisms, prompt designs tailored to specific functions in action planning, and retrievers based on diverse base models.

Third, to ensure fair comparisons, we will construct large-scale, high-quality datasets specific to the household domain and fine-tune general feature extractors on these datasets to provide a more fair and direct comparison with similarity retrieval methods.

Fourth, model transfer stability across environments within the same domain needs validation. There are significant differences when comparing the household domain with the open-ended sandbox game domain and the web domain, including variations in task nature, operational frequency, object morphology, and textual input complexity. Therefore, we will train retrievers for the web, open-ended sandbox game, and household domains, respectively, to evaluate their generalization capability in novel environments and scenarios within each domain.

Lastly, the experimental scenarios in this study are somewhat limited, as we have only conducted experiments in the household domain. In future work, we plan to extend our approach to the web and open-ended sandbox game domains by collecting preference data through interactions and training retrievers for each domain. Subsequently, we will evaluate the generalization capabilities within the same domain, but in unseen environments and scenarios.

ACKNOWLEDGMENTS

This work was supported by NSFC under Grant 62450001 and 62476008. The authors would like to thank the anonymous reviewers for their valuable comments and advice.

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

# A    IMPLEMENTATION DETAILS

In this section, we provide more implementation details about the model, training process and implementation pipeline.

## A.1    MODEL AND TRAINING DETAILS

We transform a generative language model (MLLM) into a trajectory scoring model by replacing the language model head with a Bradley-Terry score head. In particular, both the original language model head and our proposed Bradley-Terry scoring head are single-layer MLPs. However, there are notable differences: the language model head processes all hidden states as input and generates a probability distribution over the vocabulary for each token, facilitating token sequence generation through sampling. In contrast, our Bradley-Terry scoring head relies solely on the last non-zero hidden state as input and outputs a single floating-point score. Using this approach, our model generates only one new token at a time (i.e., by setting 'max_new_tokens' to 1). In comparison to conventional MLLM training, which generates hundreds or even thousands of new tokens per iteration, our model's training is significantly more computationally efficient. All the code for the MLLM retriever model, training process, benchmark tasks and simulator modifications is available at https://github.com/PKU-RL/MART.

During training, we firstly fine-tune LLaVA to enable it to understand multiple images, details in E.1. After that, we replace the language model head with the Bradley-Terry score head, and fine-tune the model with lora. The parameter settings are listed in 6.

## A.2    DETAILS OF IMPLEMENTATION PIPELINE

In this section, we provide more details about the whole pipeline of implementation, including the data collection and retrieval in downstream tasks. The implementation pipeline is as follows:

1. Construct memory databases containing expert trajectories via the planner-based method, details in 3.2.

2. Collect the pairwise comparison data via interactive feedback to train the retriever model.

   (a) Specifically, for each task in the training set, we sample $K$ trajectories from the training memory $\mathcal{M}^{\text{train}}$, and feed them as prompts for MLLM agent to execute the embodied task respectively.

   (b) After that, based on the induced success rates of task execution, we can get the effectiveness of each trajectory for the embodied task. We can then obtain a partial order list based on success rate comparisons, producing $\binom{K}{2}$ pairs through pairwise comparison, where the trajectory with a higher success rate is treated as the positive item, and the one with a lower success rate as the negative item.

   (c) In this way, these preference pairs from interactive feedback are arranged as a positive-negative pair dataset $D$, which we use to fine-tune the MLLM according to the Bradley-Terry (Bradley & Terry, 1952) Reward Modeling loss to enhance its critiquing ability, as in Equation 1. Details are listed in Appendix 3.3 line 249-259.

3. Train the modified retriever model using the preference data. Details are listed in A.1.

4. Evaluation on unseen tasks. For each unseen task, we first retrieve trajectory with highest score for current task. Then the retrieved trajectory will be simplified through Trajectory Abstraction module. After that, the MLLM agent will execute the task with the help of abstraction of retrieved trajectory. Details are listed in Algorithm 1.

# B    EXTENSION OF RELATED WORKS

## B.1    EMBODIED GROUNDING

Grounding is a critical challenge in embodied agents, referring to the alignment between the agent and its environment.

The grounding problem can be categorized into visual grounding and embodied grounding. Visual grounding (Lai et al., 2024; Kazemzadeh et al., 2014; Nagaraja et al., 2016) addresses the problem at the perception level by identifying the most relevant object or region in an image based on a language query, whereas embodied grounding focuses on the effects of actions on environmental dynamics and how an agent generates action sequences to accomplish a given task. Approaches to addressing embodied grounding can be categorized into the following types:

**1. RL:** Reinforcement learning (RL) trains an agent's policy through interaction with the environment, making the agent inherently grounded in the environment, such as PPO (Schulman et al., 2017) and SAC (Haarnoja et al., 2018). However, RL typically requires extensive interaction with environments and often suffers from instability, making it unsuitable for MLLMs.

**2. VLA:** These methods focus on fine-tuning vision-language models (VLMs) using expert datasets collected from embodied environments, such as PaLM-E (Driess et al., 2023) and RT-2 (Brohan et al., 2023). These methods demand a significant amount of high-quality trajectory data for training.

**3. LLM as Planner:** These methods leverage Large Language Models (LLMs) or Multimodal Large Language Models (MLLMs) to generate high-level plans, which are then translated into executable action sequences by low-level controllers, such as LLM-Planner (Song et al., 2023) and P-RAG (Xu et al., 2024). A key limitation of these methods is their reliance on a predefined skill library, which restricts the scope of the agent's capabilities. Besides, acquiring a skill library might require additional RL or IL training or prior knowledge about the environment (Lifshitz et al., 2023; Yuan et al., 2024).

**4. Retrieval-Augmented MLLM Agent:** This category involves integrating task trajectory data into the prompts provided to MLLMs, such as RAP (Kagaya et al., 2024). These trajectory data, rich in grounding information about the environment, enable agents to perform tasks effectively. Retrieval-augmented methods usually demonstrate greater sample efficiency compared to RL and VLA, thanks to the use of an explicit memory buffer. Our work falls into this category.

### B.2 MULTI-MODAL INFORMATION RETRIEVAL

Recent advances in multimodal retrieval have developed various methods for encoding, fusing, and measuring similarities across different modalities. ViLT (Kim et al., 2021) directly embeds image patches with text using a Transformer, while ALIGN (Li et al., 2021) and MURAL (Jain et al., 2021) use dual-encoder architectures with EfficientNet and BERT to align modalities through contrastive learning. IMAGEBIND (Girdhar et al., 2023) extends this by creating joint embeddings for six modalities, using ViT and Transformer models. Furthermore, (Changpinyo et al., 2021) integrates users' mouse trace interactions for refined image retrieval, while ReViz (Luo et al., 2023) employs advanced encoding mechanisms for visual question answering. Building on these encoding strategies, retrieval augmentation further enhances multimodal generation (*e.g.,* RAG (Lewis et al., 2020)). FLMR (Lin et al., 2023b) addresses RA-VQA limitations by combining multi-dimensional embeddings from ColBERTv2 and ViT-based models for accurate knowledge retrieval. RA-CM3 (Yasunaga et al., 2023) enhances image captioning and text-to-image generation by using a pre-trained CLIP model to augment inputs for a CM3 Transformer. Similarly, UniRAG (Sharify-moghaddam et al., 2024) integrates retrievals using UniIR's (Wei et al., 2023) CLIP Score Fusion and BLIP Feature Fusion, improving performance in MLLMs like LLaVA. Lastly, MuRAG (Chen et al., 2022b) introduces a retrieval-augmented transformer for KB-VQA, employing T5 and ViT for multimodal encoding and retrieval from a large-scale memory bank. In contrast, our method prioritizes the effectiveness of retrieved information by employing interactive learning, ensuring that the information contributes directly to task completion.

## C EXTENSION EXPERIMENTS

We also evaluate our method on ReALFRED (Kim et al., 2025) environments, which provides realistic 3D-captured and multi-room scenes, as shown in Figure 7. The action space is similar with AI2-THOR, including fine-grained movement actions, e.g. 'move ahead', 'turn left/right x degrees', 'look up/down', and interactive atomic action including 'pick up object A', 'put object A on/in object B', 'open/close object A', 'toggle on/off object A'. Since completing tasks often involves navigating across rooms, and the scenes within these rooms closely resemble real-world environments, this

setting offers a **more diverse and challenging scenario**. The chosen task type is 'pick_and_place', which requires the agent to first navigate to the target object, pick it up, and then transport it to the designated location for placement. There are 30 tasks comprising a total 60 sub-tasks in training set, and 20 tasks including 40 sub-tasks in testing set.

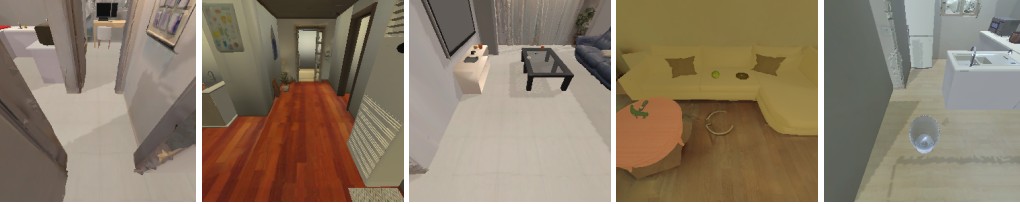

Figure 7: Image examples of ReALFRED.

The experimental results demonstrate the effectiveness of our approach compared to the baselines. As shown in Table 5, **MART** surpasses all baselines 10% in Success Rate, and reaches best performance across all metrics.

Table 5: Performance comparison of different methods in ReALFRED.

|  | PA | LP | SL | RAP | MART |
|---|---|---|---|---|---|
| **SR** ↑ | 0.25 | 0.20 | 0.26 | 0.27 | **0.37** |
| **SR-Sub** ↑ | 0.50 | 0.44 | 0.52 | 0.53 | **0.58** |
| **AS** ↓ | 101.70 | 87.93 | 89.79 | 87.89 | **78.48** |
| **AS-Sub** ↓ | 28.43 | 24.32 | 24.84 | 24.31 | **21.71** |

## D    DETAILED CASE STUDY

We present more detailed case studies, encompassing both success and failure cases for each method, along with simplified reasoning processes for clarity.

In the first detailed case, as shown in Figure 8, the MART Retriever successfully retrieved a trajectory containing the target object's location, while Trajectory Abstraction effectively compressed a 73-step trajectory into only 5 significant milestones, preserving crucial information. For the successful trial, the agent identified the target object's location (the plate on the table) through the retrieved trajectory. After some exploration, it successfully found the target object and completed the task. For the unsuccessful trial, during exploration, the agent made mistakes (highlighted in purple) and failed to complete the task within the step limit.

In the second detailed case, as shown in Figure 9, the similarity-based retriever retrieved a trajectory that appeared similar but lacked useful information. As a result, the agent had to explore independently. By chance, the agent located the target object, but the overall success rate remained low.

## E    EXPERIMENTAL SETUP DETAILS

In this appendix, we provide more low-level details on the implementation of **MART** experiments.

### E.1    MULTIPLE IMAGE INPUT IN LLAVA

The LLaVA architecture itself is compatible with using multiple images as input, but the released model weights do not have the ability to handle multiple images. In other words, when you input multiple images, it will only focus on the contents of the first image. Therefore, we fine-tune LLaVA through multi-image captioning data to enable it to understand multiple images.

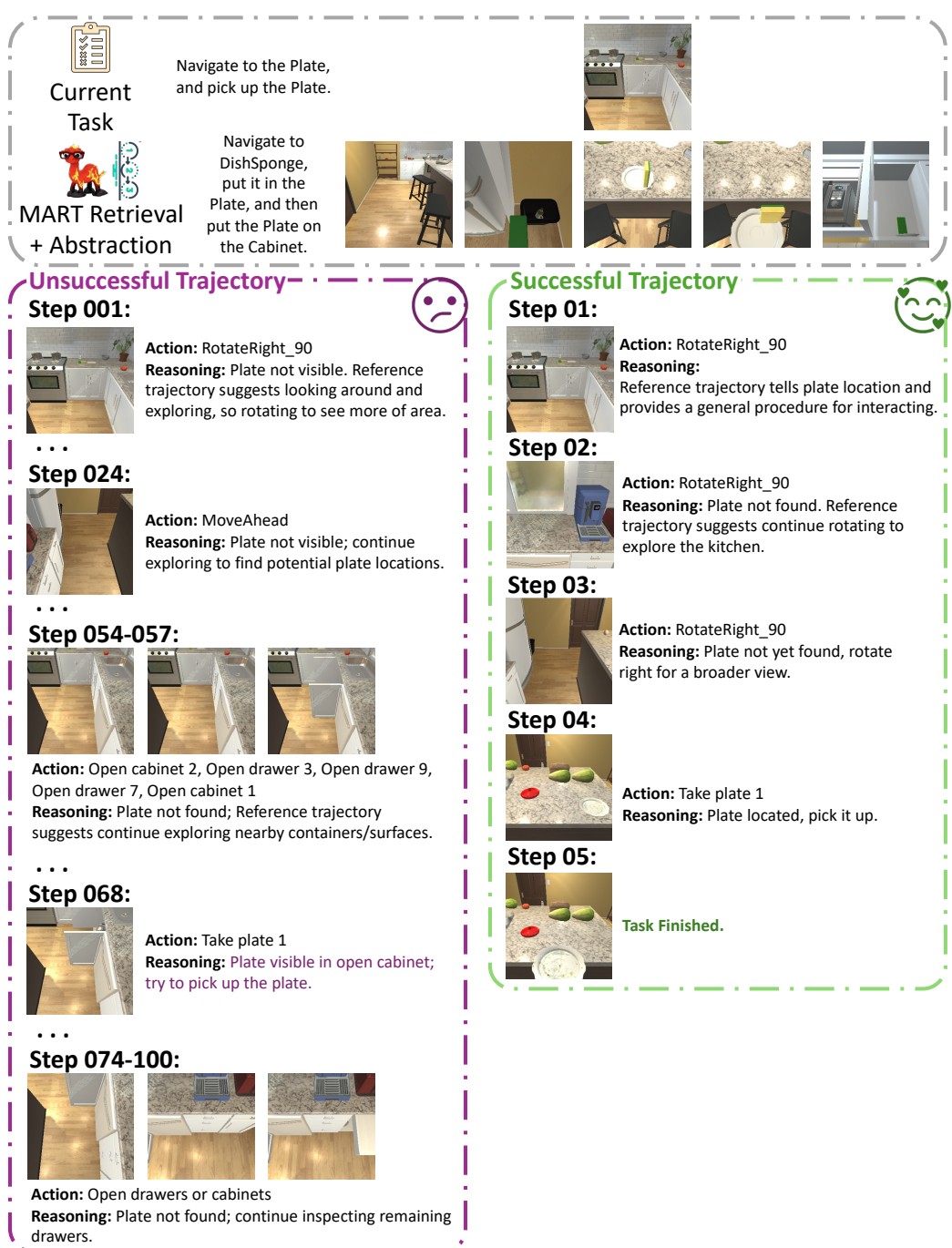

Figure 8: Comparison between unsuccessful and successful trajectory with **MART**.

We utilized LLaVA's single-image perception capabilities to enhance its multi-image perception. Specifically, we sampled images collected from the environments and employed the pre-trained LLaVA to describe the content of each image. We organized the responses into multi-image captioning data, which we then used to fine-tune LLaVA. After our verification, the fine-tuned LLaVA demonstrated the ability to perceive multiple input images. After fine-tuning with multiple image datasets, LLaVA serves as the basez model for our subsequent retriever training.

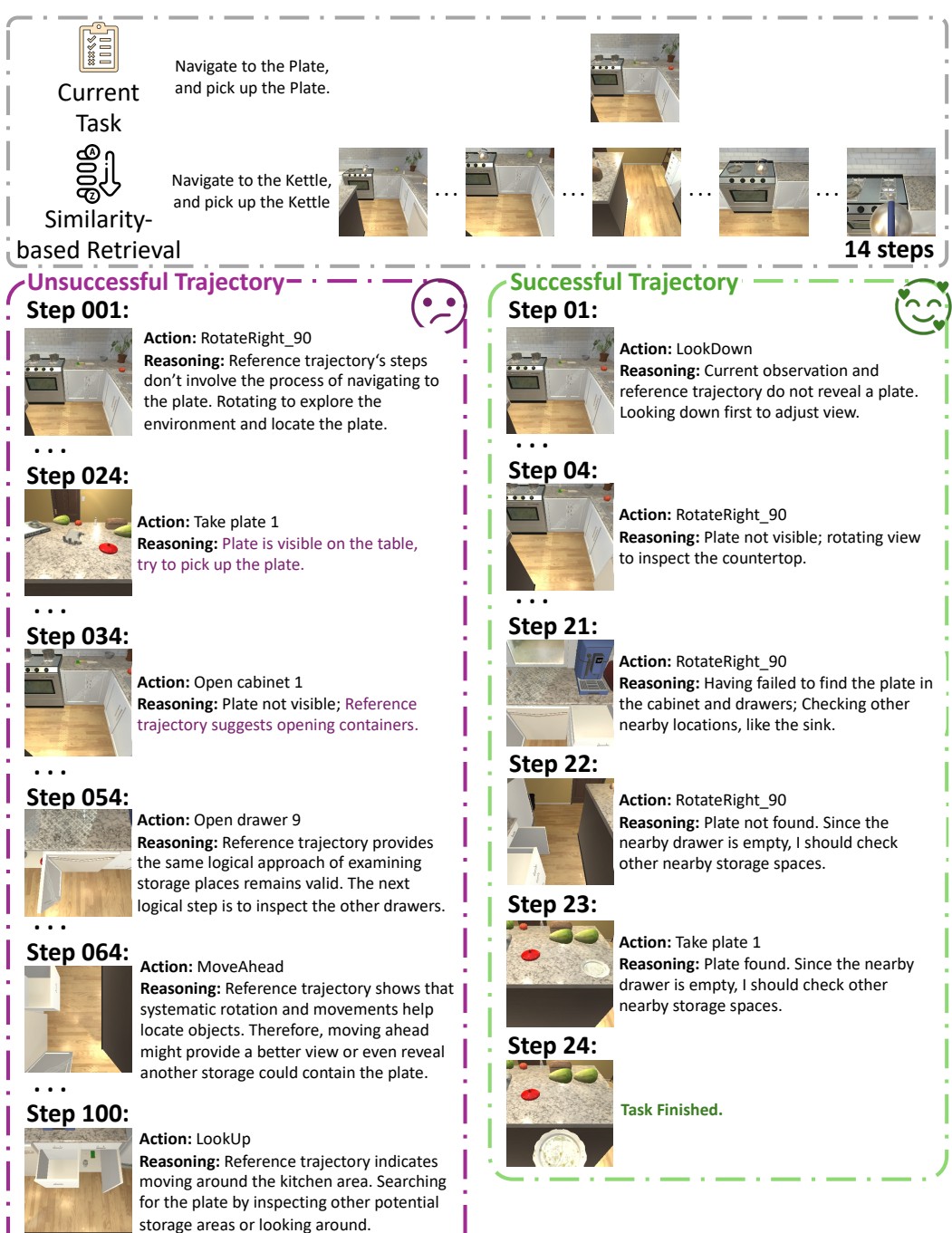

Figure 9: Comparison between unsuccessful and successful trajectory with similarity-based retrieval.

## E.2 PARAMETER SETTINGS

The specific parameter settings in the experiment are shown in Table 6.

Table 6: Hyperparameters of LLaVA fine-tuned by LoRA

| Hyperparameters | Value |
|---|---|
| LLaVA_version | llava-v1.6-mistral-7b |
| train_batch_size | 32 |
| eval_batch_size | 8 |
| gradient_accumulation_steps | 8 |
| learning_rate_AI2THOR | 2e-5 |
| mm_projector_lr_AI2THOR | 2e-5 |
| learning_rate_LEGENT | 3e-6 |
| mm_projector_lr_LEGENT | 3e-6 |
| lora_r | 16 |
| lora_alpha | 32 |
| warmup_ratio | 0.05 |
| model_max_length | 32768 |
| lr_scheduler_type | cosine |
| vision_tower | clip-vit-large-patch14-336 |

### E.3 DETAILS OF LLAVA-PLAIN

Building on existing work (Asai et al., 2024) (Sun et al., 2023b) that employs LLM for text retrieval, we use the generation probability of a special token to represent the score for LLaVA-Plain. In detail, the effectiveness score $s_i$ is measured by the probability of LLaVA-Plain to generate the special token 'Yes' and 'No', as in Equation 2, where $p\,(\text{Yes}/\text{No})$ denoted the probability of LLaVA-Plain to generate Yes or No. I.

$$s_i = \frac{p\,(\text{Yes})}{p\,(\text{Yes}) + p\,(\text{No})} \tag{2}$$

In detail, the prompts we use for LLaVA-Plain and **MART** are presented in prompt 1 of Appendix I.

## F  AI2-THOR ENVIRONMENT SPECIFICS

### F.1 SETTING DIFFERENCES

Two popular benchmarks, ALFRED (Shridhar et al., 2020), and ALFWorld (Shridhar et al., 2021) – derived from ALFRED – are both built on AI2-THOR. However, none of them are directly suitable for benchmarking MLLM agents performing real-world tasks.

ALFRED is a multimodal benchmark in AI2-THOR that uses fine-grained navigation actions. However, it requires pixel-level masks to specify objects for interaction actions. MLLM lacks the capability to generate such pixel-level masks, making ALFRED incompatible with MLLM agents without adaptation at either side.

ALFWorld adapts ALFRED for LLM agents (*i.e.,* text-only) by simplifying it. Firstly, it provides text feedback as observation, detailing objects in the agent's field of view along with their corresponding IDs. Secondly, it simplifies the action space by replacing all navigation actions with the teleportation action 'go to' and composite high-level actions like 'heat', 'clean', and 'cool', each involving multiple atomic interactions. For example, the "cool object a" action is equivalent to: 'open the refrigerator', 'put object A inside the refrigerator', 'close the refrigerator', 'open the refrigerator', and 'pick up the object A'. These modifications significantly reduce task difficulty, while allowing LLM agents to perform in ALFWorld.

To better evaluate the fine-grained control abilities of MLLM agents in real-world tasks and longer-horizon more realistic tasks, we reject ALFWorld's setting approach, which uses teleportation and composite high-level actions, opting instead for fine-grained navigation actions and finer-grained actions. However, unlike in ALFRED, since MLLMs cannot generate pixel-level masks by default,

we allow interaction actions to reference objects using a numerical ID (*e.g.,* cup 1) provided by the environment's feedback, instead of a pixel-level mask.

## F.2 TASK DECOMPOSITION

We adopt the ALFRed (Shridhar et al., 2020) method to decompose the entire task into multiple sub-tasks. In ALFRed, tasks are decomposed as follows: First, they encode agent and object states, along with high-level environment dynamics, into Planning Domain Definition Language (PDDL) rules (Aeronautiques et al., 1998). Next, they define task-specific PDDL goal conditions, such as a heated potato resting on a tabletop. The planner assumes a fully observable environment with perfect knowledge of world dynamics. Consequently, each task is decomposed into several sub-tasks, with instructions provided for each subtask by human labelers.

We follow the ALFRed method but adjust the decomposition results. PDDL-based decomposition can result in inconsistent sub-task difficulty. For example, a sub-task like 'pick up object A' or 'put object A on object B', which often follows a navigation sub-task. If the previous navigation sub-task is executed successfully, the current sub-task can be completed in one step according to the PDDL decomposition. We adjusted the task decomposition by merging sub-tasks that can be completed in one step with adjacent sub-tasks to balance their difficulty. We will release the benchmark including our tasks and modified sub-tasks.

## F.3 SUCCESS DETECTION

In AI2-THOR, we provide the agent with the environment's metadata and feedback in the Success Detection module. After each step is executed, the agent then determines whether the current sub-task is completed using a few-shot approach. For specific prompts, please refer to Appendix I.

## F.4 HIERARCHY EXAMPLES

Figure 10 illustrates multiple cases of the object hierarchy relationship *Inside*, only available in the AI2-THOR environment.

Moreover, Figure 11 illustrates the different types of rooms in our LEGENT experiments, showing connectivity, but simpler *On* hierarchies than AI2-THOR.

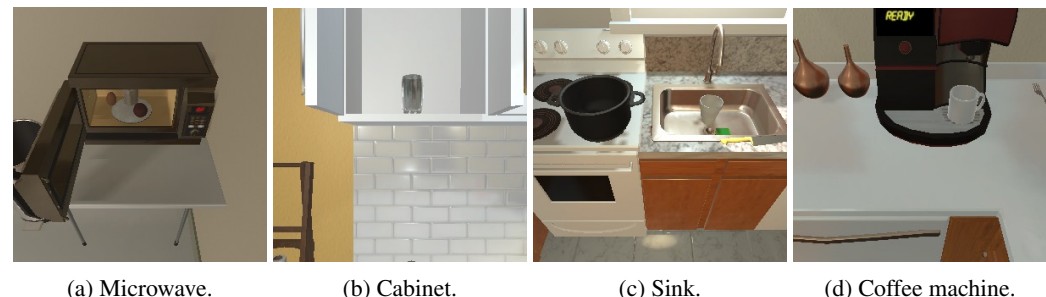

(a) Microwave.    (b) Cabinet.    (c) Sink.    (d) Coffee machine.

Figure 10: Image examples of object hierarchy in AI2-THOR.

## G   FULL STEP COUNT RESULTS

Due to space limitations, we present the full tables with Success Rate (SR) and Average Steps (AS) results here.

Table 7 shows the performance comparison of different types in LEGENT, with full step count results. While, Table 8 shows the performance comparisons of the ablation studies in both AI2-THOR and LEGENT environments, with full step count results.

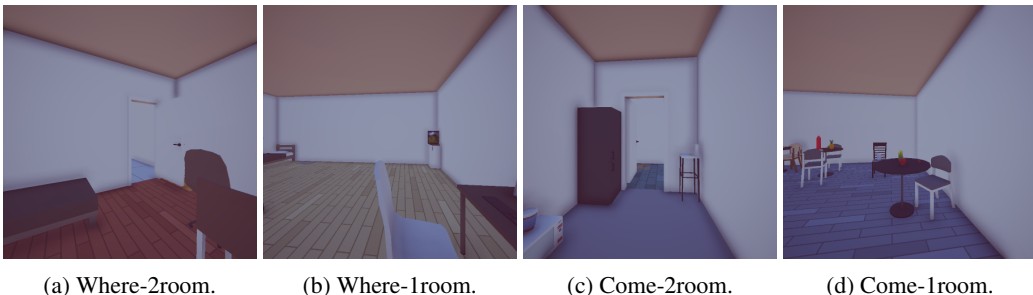

(a) Where-2room.  (b) Where-1room.  (c) Come-2room.  (d) Come-1room.

Figure 11: Image examples of different types of tasks in LEGENT.

Table 7: Performance comparison of different types in LEGENT, with full step count results.

| Metric | Type | PA | LP | SL | RAP | MART |
|---|---|---|---|---|---|---|
| **SR ↑** | Where-2room | 0.65 | 0.60 | 0.60 | 0.73 | **0.88** |
| | Where-1room | 0.78 | 0.80 | 0.80 | 0.89 | **0.91** |
| | Come-2room | 0.63 | 0.55 | 0.63 | 0.46 | **0.73** |
| | Come-1room | 0.75 | 0.82 | 0.96 | 0.93 | **0.98** |
| | **Average** | 0.70 | 0.69 | 0.75 | 0.75 | **0.87** |
| **AS ↓** | Where-2room | 25.23 | 28.98 | 28.30 | 26.23 | **14.03** |
| | Where-1room | 10.40 | 16.49 | 15.36 | 9.82 | **5.07** |
| | Come-2room | 32.05 | 34.95 | 28.80 | 35.33 | **26.90** |
| | Come-1room | 26.80 | 19.62 | 11.22 | 11.09 | **9.24** |
| | **Average** | 23.62 | 25.01 | 20.92 | 20.62 | **13.81** |

## H ALGORITHM

**MART**'s Agent Execution Pseudocode is shown in Algorithm 1.

---

**algorithm 1 MART** Agent Execution Pseudocode

---

**Input:** Expert Trajectory Memory $\mathcal{M}$, Retriever $q_\theta$, Policy $\pi$, Task $\ell^c$, Horizon $H$, Initial Observation $o_1^c$, Preference Pairs $\mathcal{D}$
**Output:** the Success status of task execution
1: Fine-tune retriever $q_\theta$ with Preference Pairs $\mathcal{D}$
2: Retrieve reference trajectory $\tau^e \leftarrow q_\theta(\ell^c, o_1^c, \mathcal{M})$
3: *TrajectoryAbstraction* to simplify $\tau^e$
4: **for** $t = 1$ to $H$ **do**
5:     **if** $t! = 1$ **then**
6:         $r_t \leftarrow SelfReflection(a_{t-1}, o_{t-1}^c, f_{t-1}^c)$
7:         Select action $a_t \leftarrow \pi(\ell^c, \tau^e, o_t^c, r_t)$
8:     **end if**
9:     **if** $t == 1$ **then**
10:         Select action $a_t \leftarrow \pi(\ell^c, \tau^e, o_t^c)$
11:     **end if**
12:     $(o_t^c, f_t^c) \leftarrow ActionExecution(a_t)$
13:     **if** $SuccessDetection(\ell^c, f_{t+1}^c)$ **then**
14:         **return True**   // Task successfully completed
15:     **else if** $t \geq H$ **then**
16:         **return False**   // Task failed after reaching horizon
17:     **end if**
18: **end for**

---

Table 8: Ablation studies of **MART** in the AI2-THOR and LEGENT environments, with full step count results.

| Environment | Metric | | w/o Abstraction | Sim.+FTM | MART |
|---|---|---|---|---|---|
| **AI2-THOR** | **SR ↑** | | 0.31 | 0.34 | **0.40** |
| | **SR-Sub ↑** | | 0.73 | 0.74 | **0.75** |
| | **AS ↓** | | 81.22 | 78.09 | **78.48** |
| | **AS-Sub ↓** | | 22.47 | 21.60 | **21.71** |
| **LEGENT** | **SR ↑** | Where-2room | 0.78 | 0.72 | **0.88** |
| | | Where-1room | 0.74 | 0.89 | **0.91** |
| | | Come-2room | 0.63 | 0.55 | **0.73** |
| | | Come-1room | 0.95 | 0.91 | **0.98** |
| | | **Average** | 0.77 | 0.77 | **0.87** |
| | **AS ↓** | Where-2room | 17.36 | 15.96 | 14.03 |
| | | Where-1room | 12.36 | 11.51 | 5.07 |
| | | Come-2room | 32.75 | 34.13 | 26.90 |
| | | Come-1room | 11.47 | 13.73 | 9.24 |
| | | **Average** | 18.48 | 18.83 | 13.81 |

# I  AGENT PROMPTS

Prompt 1: Retriver Prompt to LLaVA.

```
You are a highly intelligent vision language assistant agent situated in
    a virtual environment.
You have been given a task instruction that you need to complete.
Additionally, you are provided with a reference trajectory, which
    includes previous task instructions, actions, and egocentric
    observations from the same virtual environment.
This reference trajectory represents a successful completion of a task
    and is intended to guide you in performing the current task.
The current task instruction is: [current task].
The task instruction of reference trajectory is [memory task].
Among these input images, the 1st image is your current observation,
    while the other images are milestones of observations from the
    reference trajectory.
The abstraction of the reference trajectory is:

<Description of Milestone 0>, <Image 0>, <Feedback 0>, <Action 0>;
<Description of Milestone 1>, <Image 1>, <Feedback 1>, <Action 1>;
...

Your should thoroughly understand the current task and the reference
    trajectory. Then, analyze whether the reference trajectory can assist
     in executing the current task by answering 'Yes' or 'No'.
You should only respond in the format described below, and you should not
     output comments or other information:
Answer: Yes or No.
```

Prompt 2: Trajectory Abstraction Prompt.

```
You are a highly intelligent vision-language assistant agent placed
    within a virtual environment. You are provided with a trajectory
    consisting of:
- Trajectory Task Instruction: {Task Instruction}
- A sequence of first-person perspective observations (<Image x>)
- A sequence of environment feedbacks (<Feedback x>)
- A sequence of actions (<Action x>)
- Another Task Instruction: {Current Task Instruction}

Your Tasks:
1. Fully comprehend the tasks accomplished in the trajectory.
2. Identify the significant milestones in the trajectory that are
    essential for accomplishing the task. These are points where
    important decisions are made, goals are achieved, or notable changes
    in the environment or state occur. Do not treat every image as a
    significant milestone. If the target object of Another Task
    Instruction appears in the trajectory feedback, then the point where
    it appears is also a significant milestone because it is very
    important to Another Task Instruction.
3. For each significant milestone, provide:
- A description of the milestone.
- The corresponding image (<Image x>).
- The corresponding feedback (<Feedback x>).
- The sequence of actions taken between this milestone and the next one.

<Image 0>, <Feedback 0>, <Action 0>;
<Image 1>, <Feedback 1>, <Action 1>;
...

Response Format (do not include any comments or additional information):
1. {Description of significant milestone 1}: <Image a>. <Feedback a>.
    Actions: {Actions taken between this significant milestone and the
    next significant milestone} (such as <Action a>, <Action b>).
2. {Description of significant milestone 2}: <Image c>. <Feedback c>.
    Actions: {Actions taken between this significant milestone and next
    significant milestone}
...

Notes:
- Ensure that the number of <Image x> and <Action x> matches the provided
     trajectory.
- Only include the specified information in your response.
- A significant milestone is a point in the trajectory where a key part
    of the task is accomplished. If the target object of Another Task
    Instruction appears in the trajectory feedback, then the point where
    it appears is also a significant milestone because it is very
    important to complete Another Task .
```

Prompt 3: Self-Reflection Prompt.

```
You are a vision language assistant agent with high intelligence.
You are placed inside a virtual environment, equipped to handle a wide
    range of tasks in the virtual environment.
Your advanced capabilities enable you to process and interpret egocentric
     observation screenshots and environment feedback.
Your task is to examine these inputs, interpret the environmental
    feedback, and determine whether the executed action takes effect.

Current task:
{task_instruction}

Previous actions:
{previous_action_1}, {previous_action_2}, ...

Previous environment feedback:
{previous_feedback}

Reasoning for the previous actions:
{previous_reasoning_1}, {previous_reasoning_2}, ...

Previous observations:
{previous_image_1}, {previous_image_2}, ...

Reasoning: You need to answer the following questions step by step to get
     some reasoning based on the previous actions and sequential images
    of the execution of the previous actions.
1. What is the last executed action?
2. Was the last executed action successful? Give reasons. You should
    refer to the following rules:
- If the action involves movement of position, change of view, or
    interaction with an object, it is considered unsuccessful when the
    image you currently observe remains unchanged as the previous frame.
3. If the last action is not executed successfully, what is the most
    probable cause? You should give only one cause and refer to the
    following rules:
- If it is an interaction action, the most probable cause was that the
    object id of the interaction was wrong.
- If it is a movement action, the most probable cause was that you were
    blocked by seen or unseen obstacles.
4. If the last action is executed successfully, Does the previous action
    sequence promote the progress of the current task?
5. If the answer to Reasoning Question 4 is No, how should you adjust to
    promote the progress of the current task?

You should only respond in the format described below, and you should not
     output comments or other information:
Reasoning:
1. ...
2. ...
3. ...
4. ...
5. ...
...
```

Prompt 4: Action Planning Prompt.

```
You are a robot working in a household environment. You can move and
    interact with the objects you see.
The actions you can perform include:
1. 'MoveAhead': Move one step forward.
2. 'RotateLeft_$degree': Turn to the left by the specified number of
    degrees, ranging from 0 to 180 degrees.
3. 'RotateRight_$degree': Turn to the right by the specified number of
    degrees, ranging from 0 to 180 degrees.
4. 'LookUp': Look up 30 degrees.
5. 'LookDown': Look down 30 degrees.
6. 'Take $objectID': You can take any object in your line of sight. The
    $objectID can be obtained from the environment feedback.
7. 'Put $objectID on/in $targetID': You can put the object in your hand
    onto/into the target receptacle. The $targetID can be obtained from
    the environment feedback, and $objectID refers to the object in your
    hand.
8. 'Open $objectID': You can open any openable object in your line of
    sight. The $objectID can be obtained from the environment feedback.
9. 'Close $objectID': you can close any open object in your line of sight
    . The $objectID can be obtained from the environment feedback.
10. 'ToggleOn $objectID': You can toggle on the switch of the object,
    such as a faucet or microwave. The $objectID can be obtained from the
     environment feedback.
11. 'ToggleOff $objectID': You can toggle off the switch of the object,
    such as faucet or microwave. The $objectID can be obtained from the
    environment feedback.
12. 'Slice $objectID': You can slice any object in your line of sight.
    The $objectID can be obtained from the environment feedback.

Examples of actions: RotateLeft_30; MoveAhead; Take mug 1; Open fridge 1;
     ToggleOn microwave 1; Close fridge 1;...

You need to follow the task instructions to complete the task. Here is
    some helpful information.

Current task:
{task}

Current environment feedback:
{obs}

Previous environment feedback:
{previous_obs}

Previous action and reasoning:
{previous_action}

Current observation
<image>

Previous observation
<image>

Reference trajectory abstraction: The reference trajectory is a
    successful trajectory, which is used to guide you to complete the
    current task. The task of reference trajectory is {current task}.

<Description of Milestone 0>, <Image 0>, <Feedback 0>, <Action 0>;
<Description of Milestone 1>, <Image 1>, <Feedback 1>, <Action 1>;
...

Based on the above information, you should first analyze the current
    situation, and provide the reasoning for what you should do for the
```

```
    next step to complete the task. Then, you should output the exact
    action you want to execute in the simulator.

Reasoning: You should think step by step and provide detailed reasoning
    to determine the next action executed on the current state of the
    task. You need to answer the following questions step by step.

1. Does reference trajectory abstraction exist? If the answer is no,
    ignore the questions from number 2 to number 4.
2. What process does reference trajectory abstraction describe?
3. Consider what is your current task. Based on the Observation of the
    previous step and Current Observation, which waypoint has the current
     task reached?
4. Based on the answer of the question number 3, you should consider how
    the current waypoint and the parts after that in the reference
    trajectory abstraction can help you with your current task. The help
    provided by the reference trajectory abstraction can be knowing the
    location of the target object or knowing the execution flow of a
    combined action.
5. Based on the completion progress of the current task and the answer to
     question number 4, what should you do for the next step?
6. Why do you take this action next step?

Action: The best action to execute next to progress in completing the
    task. You should pay more attention to the following action rules:
1. Given the current situation and task, you should only choose the most
    suitable action from the valid action set. You cannot use actions
    that are not in the valid action set to control the application,
    especially 'Await Next Task'.
2. If the Action of the previous step fails, you should not continue
    trying but should consider adjusting your position to get closer to
    the target object.
3. You MUST NOT match or imitate the reference trajectory. You should
    think about how to complete the current task based on the answer to
    Reasoning Question 4.

You should only respond in the format described below, and you should not
     output comments or other information:
Reasoning:
1. ...
2. ...
3. ...
4. ...
5. ...
6. ...

Action: ...
```

Prompt 5: Success Detection Prompt.

```
You are a highly intelligent vision-language assistant agent.
You are situated in a virtual environment, equipped to handle a diverse
    array of tasks.
Your advanced capabilities allow you to process and interpret egocentric
    observation screenshots, environmental feedback and environmental
    metadata.
Your task is to examine these inputs, understand the environmental
    metadata, and assess the success of the current task.

Current task:
<task>

Environmental Metadata
<environment metadata>

Environmental Feedback
<environment feedback>

Current Inventory
<inventory>

You need to refer to the following rules:

1. If the current task contains multiple tasks, it is considered
    successful only when each task succeeds.
2. If the task is a navigation task, you need to check the environmental
    metadata. The navigation task succeeds when the target object is in
    view and the distance is less than 1m.
3. If the task is a pickup task, then according to the environmental
    feedback, the pickup task succeeds when the target object is in your
    inventory.
4. If the task is a put down task (put object a on/in object b), the put
    down task succeeds when the environmental feedback from the
    environment includes You put object a on/in the object b successfully
    .
5. If the task is a clean task, the clean task succeeds when the
    cleaned_objects in environmental metadata include the target object
    and the same target object is also in inventory.
6. If the task is a cool task, the cool task succeeds when the
    cooled_objects in environmental metadata includes the target object
    and the same target object is also in inventory.
7. If the task is a heat task, the heat task succeeds when the
    heated_objects in environmental metadata includes the target object
    and the same target object is also in inventory.

You should only respond in the format described below, and you should not
output comments or other information:

Answer: True or False.
```

