# OpenReview forum: "MLLM as Retriever: Interactively Learning Multimodal Retrieval for Embodied Agents"
_ICLR.cc/2025/Conference — ICLR 2025 Poster_

### Official Review · Reviewer_7byC · 2024-10-21

**Soundness:** 2
**Presentation:** 3
**Contribution:** 2
**Rating:** 6
**Confidence:** 5

**Summary:**

The paper introduces MLLM As Retriever (MART), a novel approach designed to enhance the performance of embodied agents in complex tasks by improving multimodal retrieval of task-relevant trajectory data. Traditional retrieval methods often rely on surface-level similarities of textual or visual cues, which may not effectively capture the nuances required for specific tasks. MART addresses this limitation by fine-tuning a Multimodal Large Language Model (MLLM) retriever through interactive preference learning, allowing it to evaluate and prioritize trajectories based on their effectiveness for unseen tasks. Additionally, the paper presents Trajectory Abstraction, a mechanism that leverages MLLMs’ summarization capabilities to represent trajectories with fewer tokens while preserving essential information, enabling agents to better understand key milestones. Experimental results across various environments demonstrate that MART significantly improves task success rates on unseen tasks compared to baseline methods, consistently surpassing them by over 10%. The authors commit to releasing all benchmark task sets and simulator code modifications to facilitate further research.

**Strengths:**

1. Innovative Retrieval Method: The paper introduces a novel method by fine-tuning an MLLM as a retriever using interactive preference learning, moving beyond traditional similarity-based retrieval methods. This allows the retriever to assess trajectory effectiveness more holistically.
2. Trajectory Abstraction Mechanism: The introduction of Trajectory Abstraction is a significant contribution. By condensing trajectories into shorter representations while retaining crucial information, agents can process and utilize past experiences more efficiently.
3. Demonstrated Performance Improvements: The method substantially improves task success rates across various environments and unseen tasks, indicating its effectiveness and potential for broader applications in embodied AI.
4. Advancement in Embodied Agent Capabilities: By enhancing how agents retrieve and utilize past trajectories, MART contributes to developing more intelligent and capable embodied agents capable of handling complex, long-horizon tasks.
5. Commitment to Open Science: The release of benchmark task sets and simulator code modifications promotes transparency and allows other researchers to build upon this work, fostering collaboration in the community.

**Weaknesses:**

1. Efficiency Issues: The proposed method of using a Multimodal Large Language Model (MLLM) as a retriever may not be as efficient as conventional similarity retrieval methods, known for being fast and low-cost. Given that complex language model reasoning can already achieve high-quality planning decomposition and performance, the added cost of training and deploying an additional MLLM may not be justified. There is a need for verification to determine if the high cost is worth the potential benefits, including considerations of additional inference time and required training resources.

2. Potential Stability and Hallucination Problems: Replacing simple similarity calculations with judgments based on a language model introduces potential instability and hallucination issues inherent to language models. The MLLM’s retrieval process might be inconsistent, leading to unreliable action selection. It’s unclear how these issues are overcome in the system and whether effectiveness has been verified across multiple evaluation methods. Since MLLM retrieval effectively acts as a terminal evaluator to judge the best action, stability is critical.

3. Experimental Confusion: In Table 4, the ablation study comparing MLLM Retrieval and similarity retrieval methods shows minimal differences in Average Steps (AS) but noticeable differences in Success Rate (SR). Questions arise about the correlation between these two evaluation metrics. Specifically:

    a) Clarification of AS: What does AS (Average Steps) represent? Is it the number of iterations the agent system takes to succeed once?

    b) Difference in Metrics: Why is there a significant difference in SR but not AS between the two methods?

    c) Fairness of AS: Given the introduction of an additional language model, using AS may not be fair. It might be more reasonable to replace AS with the average inference time of the entire system to reflect the true cost and efficiency.

4. Unfair Comparison in Case Study: The case study presents success cases of the proposed MLLM Retrieval method alongside failure cases of the similarity retrieval method. This comparison may not be fair or sufficient to demonstrate the advantages of the proposed method. Providing success and failure cases for each method would offer a more balanced and comprehensive evaluation.

**Questions:**

1. Is the High Cost of Training an Additional MLLM Justified?
Could you verify whether the benefits of training and using an additional MLLM outweigh the high costs in terms of additional inference time and required training resources? Are there specific scenarios where the MLLM retriever significantly outperforms conventional similarity retrieval methods to justify its use?

2. How Did You Overcome Stability and Hallucination Issues?
Since language models can be unstable and prone to hallucinations, what methods did you employ to mitigate these issues in your MLLM retrieval system? Have you verified the effectiveness of your approach across multiple evaluation methods to ensure reliable performance?

3. Clarification on Evaluation Metrics in Table 4:

	a) What Exactly Does AS Represent? Could you clarify the meaning of AS (Average Steps) in your evaluation? Does it represent the number of iterations required for the agent to succeed once?

	b) Correlation Between AS and SR: Does Average Steps and Success Rate correlate? How do you interpret the results where AS shows minimal difference but SR differs more significantly?

	c) Fairness of Using AS as a Metric:
Considering the additional computational cost of incorporating an MLLM, would it be more appropriate to use the average inference time of the entire system instead of AS to ensure a fair comparison between methods?

4. Can You Provide Balanced Case Studies?
To ensure a fair comparison, could you provide both the success and failure cases for each method (MLLM Retrieval and similarity retrieval) in your case studies? This would help understand each approach's strengths and weaknesses more comprehensively.

---

> ### Author Response · Authors · 2024-11-25
> **Thanks for your review! Here, we respond to your comments and address the issues. We hope to hear back from you if you have further questions!**
>
> **Q1.** Whether the benefits of training and using an additional MLLM outweigh the high costs in terms of additional inference time and required training resources? Are there specific scenarios where the MLLM retriever significantly outperforms conventional similarity retrieval methods to justify its use?
>
> **A1.**
>
> > Whether the benefits of training and using an additional MLLM outweigh the high costs in terms of additional inference time and required training resources?
>
> Thanks for pointing out the efficiency issues! We have been carefully considering the issue of efficiency and striving to reduce the cost of training and evaluation. Our method is not expensive during training and inference.
>
> **Training Phase:**
>
> We replaced the original language model head of LLaVA with a Bradley-Terry score head, enabling our model to **generate only one token** at a time (i.e., by setting ‘max new tokens’ to 1). Compared to conventional MLLM training, which generates **hundreds or thousands of tokens** per iteration, this approach is significantly more computationally efficient. Additionally, we utilized **LoRA** during training to further reduce costs. Specifically, training our method required **only 8 hours** on 8 A100-40GB GPUs.
>
> **Evaluation Phase:**
>
> For evaluation, we utilized **a single 4090 GPU**. To optimize speed, we processed **multiple memory trajectories into batches for each query**, reducing the complexity to $O(n/b)$, where $n$ represents the total number of memory trajectories and $b$ represents the number of trajectories per batch.
>
> We also compared our inference time with the similarity-based method, and the results are summarized in the table.
>
> |                   | Similarity-base Retrieval | MLLM retriever | Action Planning |
> | ----------------- | ------------------------- | -------------- | --------------- |
> | Seconds per query | 0.05                      | 0.43           | 15.06           |
> | Repetitions       | 1                         | 1              | multiple        |
>
> However, for the entire pipeline, the main bottleneck is the action planning part. That is, for each task, we only perform **one retrieval at the beginning** and then perform **multiple action planning** processes to complete the task. The average time spent on **one action planning** is **15.06** seconds, and this time has to **repeat dozens of times**. Although the retrieval stage of  MART has a higher time complexity compared to the similarity-based retrieval method, the actual difference is minimal, as the primary time bottleneck lies in action planning.
>
> Considering the comprehensive experiments in **three** various environments on unseen tasks, MART **consistently** surpasses baselines by **over 10% across different environments**. So, we assume the benefits of training additional retriever model outweigh the cost of training and evaluation.

---

> > ### Comment · Reviewer_7byC · 2024-11-25
> > **Absence of a detailed ablation and potential more resources**
> >
> > It’s commendable that you’ve chosen to modify the original head of LLaVA to compress output, which likely enhances computational efficiency. However, this approach raises concerns about the additional resources needed for labeling a novel dataset, especially if direct success rate measurements are unavailable to train your MLLM to maintain accurate classification or evaluation performance.
> >
> > Furthermore, **the absence of a detailed ablation study** on planning LLM and MLM retrieval is notable. You cannot guarantee the **strict positive correlation** between the retrieval and planning processes, but the simple similarity method naturally has this feature. If the planning process is inaccurate, how do you ensure that the success rate used to label data for your retrieval model doesn’t compound errors, potentially reinforcing incorrect planning but correct retrieval scenarios?
> >
> > Additionally, it would be beneficial to understand the specific datasets utilized, the labeling process, and whether each application requires training a new model on domain-specific datasets instead of leveraging SOTA LLMs like GPT-4 or even O1.

---

> ### Author Response · Authors · 2024-11-25
> **Response to the remaining part of question 1.**
>
> > Are there specific scenarios where the MLLM retriever significantly outperforms conventional similarity retrieval methods to justify its use?
>
> We evaluated the SR across different **task types** and observed that our method **consistently** outperformed similarity-based methods on unseen tasks across diverse environments.
>
>
>
> | Environment | Task Type             | Similarity-base Retriever | MLLM Retriever |
> | ----------- | --------------------- | ------------------------- | -------------- |
> | AI2-THOR    | pick_and_place        | 0.25                      | **0.35**       |
> | AI2-THOR    | pick_clean_then_place | 0.25                      | **0.40**       |
> | AI2-THOR    | pick_cool_then_place  | 0.22                      | **0.40**       |
> | LEGENT      | Where-2room           | 0.73                      | **0.88**       |
> | LEGENT      | Where-1room           | 0.89                      | **0.91**       |
> | LEGENT      | Come-2room            | 0.46                      | **0.73**       |
> | LEGENT      | Come-1room            | 0.93                      | **0.98**       |
> | REALFRED    | pick_and_place        | 0.27                      | **0.37**       |
>
> The primary issue with similarity-based retrieval methods lies in their **vulnerability to interfering trajectories**. These trajectories, although **visually or textually similar** to the current task instructions or image observations, are irrelevant to the task and **do not provide meaningful assistance**. This interference can mislead similarity-based methods, resulting in the retrieval of trajectories that are ineffective for task completion.
>
> As demonstrated in **Case Study 1**, the similarity-based method retrieved a highly similar trajectory, which was irrelevant to the current task and offered minimal assistance, leading to a low success rate. In contrast, our method emphasizes the practical effectiveness of trajectories in completing tasks. This focus allows it to mitigate the impact of interfering trajectories and retrieve trajectories that are truly effective, thereby ensuring higher task success rates.
>
> **Figure 2** (line 67) further illustrates that relying solely on **similarity** is **ineffective** for retrieving useful trajectories, as similarity is not directly correlated with success rate. The linear fit curve demonstrates that there is virtually no correlation between similarity and the actual success rate. In contrast, the scores produced by our trained retriever model exhibit a clear positive correlation with success rate.

---

> ### Author Response · Authors · 2024-11-25
> **Response to the question 2.**
>
> **Q2.** How did you overcome and mitigate stability and hallucination issues? Have you verified the effectiveness of your approach across multiple evaluation methods to ensure reliable performance?
>
> **A2.** Thanks for the valuable suggestion!
>
> > How did you overcome and mitigate stability and hallucination issues?
>
> Since we replace the language model head with a Bradley-Terry score head, our retriever model maps the hidden states to scalar scores, **diverging from the paradigm of a language generation model**. The established **definition of hallucination in LLM doesn't exactly suit the case**.  The hallucination is **not** a topic to discuss in this scope and is also **not** included in the **prior works listed below**:
>
> Sun et al. [1] propose to use LLM for relevance ranking in IR, which use pairwise comparisons and sliding window strategy to choose the most relevant candidate. Their experiments reveal that properly instructed LLMs can deliver competitive, even **superior results to state-of-the-art** supervised methods on popular IR benchmarks.
>
> HELM [2] propose to a pointwise ranking method, which prompt LLMs to output either “Yes” or “No”, and the generation probability is used to determine the relevance of the candidates for a given query.
>
> SELF-RAG [3] trains a LM that adaptively retrieves passages on-demand, and generates and reflects on retrieved passages and its own generations using special tokens, called reflection tokens. Experiments show that SELF-RAG (7B and 13B parameters) **significantly outperforms state-of-the-art LLMs and retrieval-augmented models** on a diverse set of tasks.
>
> **Reference**
>
> [1] Is ChatGPT good at search? investigating large language models as re-ranking agents. EMNLP 2023
>
> [2] Holistic evaluation of language models, TMLR 2023
>
> [3] Self-RAG: Learning to Retrieve, Generate, and Critique through Self-Reflection, ICLR 2024 Oral
>
> In terms of potential **instability**, it is indeed a problem that the **model may output scores that contradict facts**. In our retrieval task, the instability manifests as assigning higher scores to less relevant trajectories, which is a **shared problem between both similarity-based methods and our LLM-based method**.
>
> Due to the lack of the ground truth of the relevance rank of trajectories, it's hard to assess the instability directly. Considering all the methods share the same procedure in the downstream task execution, the effectiveness of trajectory selection is directly reflected in the success rate. From this aspect, the outperformance of our methods over the retrieval-based baseline reflects the less instability of our method.
>
> Furthermore, while complete mitigation of instability remains challenging, we implement a design to mitigate its impact. In the action planning module, we prompt the agent to analyze the referenced trajectory and evaluate its contribution to the current task. When determining the referenced trajectory is useless, the agent will execute the task with solely task requirements and observations.
>
> To evaluate the effectiveness of this design, we conducted experiments where manually inputting trajectory that had low MART score and is manually verified to be irrelevant to the current task into the agent. Besides, we remove the prompt of mitigating design as an ablation experiment.
>
> The ablation results demonstrated the effectiveness of the mitigating design.
>
> |                     | Irrelevant Trajectory | Irrelevant Trajectory w/o mitigating |
> | ------------------- | --------------------- | ---------------------------------------------- |
> | SR $\uparrow$       | 0.20                  | 0.15                                          |
> | SR-Sub $\uparrow$   | 0.65                  | 0.59                                         |
> | AS $\downarrow$     | 150.30                | 168.74                                        |
> | AS-Sub $\downarrow$ | 42.25                 | 46.85                                         |
>
> > Have you verified the effectiveness of your approach across multiple evaluation methods to ensure reliable performance?
>
> After retrieval, our method and the baseline share **identical** settings in downstream tasks. Therefore, the success rate in downstream tasks **fairly** reflects whether the model can select truly effective trajectories. On this metric, our method **significantly outperforms** the retrieval-based baseline, demonstrating the effectiveness of our approach.
>
> Furthermore, if **alternative evaluation methods** were available to verify whether the effectiveness score produced by our model is accurate, these methods would show a **strong positive correlation with the success rate**. This is because the effectiveness score directly represents the **framework's capability to achieve task successfully**. Thus, we believe the success rate is a **sufficient** evaluation metric.

---

> > ### Comment · Reviewer_7byC · 2024-11-25
> > **Fairness in Method Comparison**
> >
> > Your response does not fully address my concerns about the fairness of comparing your method with “similarity retrieval methods.” The effectiveness of similarity retrieval largely depends on the quality of representation or feature extraction. If the **features extracted can accurately** represent and adapt to the shifts in your domain, similarity retrieval could be highly efficient. This returns to my earlier point about constructing high-quality and voluminous datasets to generalize across your proposed diverse scenarios. Could a more straightforward method exist than building such expansive datasets, particularly when domain-specific training seems essential?

---

> > > ### Comment · Reviewer_7byC · 2024-11-25
> > > **Concerns Regarding Hallucinations and Stability**
> > >
> > > While transitioning from a language model head to a score head is a strategic move to reduce language hallucinations, this change doesn’t fully address the underlying instability issues. The scenarios you’ve tested appear similar, which does not convincingly demonstrate the model’s stability in out-of-distribution (OOD) situations. This could lead to severe problems when the model encounters truly novel or unexpected environments.

---

> > > > ### Comment · Reviewer_7byC · 2024-11-25
> > > > **Inclusion of Diverse Tasks**
> > > >
> > > > While your approach is insightful, incorporating a broader range of tasks could enhance its robustness and applicability. Tasks from different domains, such as webArena, textcraft, or even directly from Minecraft, could demonstrate the model’s effectiveness across a variety of scenarios, making your methodology appear more solid and widely applicable.

---

> > > > > ### Author Response · Authors · 2024-11-27
> > > > >
> > > > > We sincerely thank you for your insightful suggestion to broaden our experimental scenarios!
> > > > >
> > > > > We appreciate the suggestion to broaden our experimental scenarios. However, due to time constraints, completing this within the rebuttal period is highly challenging.
> > > > >
> > > > > As we previously stated, there are significant differences when comparing the household domain with the open-ended sandbox game domain and the web domain, including variations in task nature, operational frequency, object morphology, and textual input complexity.
> > > > >
> > > > > Therefore, in future work, we plan to extend our approach to the web and open-ended sandbox game domains by collecting preference data through interactions and training retrievers for each domain. Subsequently, we will evaluate the generalization capabilities within the same domain, but in unseen environments and scenarios.

---

> > > > ### Author Response · Authors · 2024-11-27
> > > >
> > > > We sincerely appreciate your insightful suggestion!
> > > >
> > > > First and foremost, it is crucial to clarify that our approach does not focus on cross-domain generalization. In each environment, preference data must be collected through interactions within the environment to train the model. Out-of-distribution (OOD) scenarios **fall outside the scope of our method** and are highly challenging due to the **significant gaps** between different domains.
> > > >
> > > > There are substantial differences between the **household** domain and the **open-ended sandbox game** domain:
> > > >
> > > > - In terms of **task nature**, tasks in the household domain involve static object manipulation and navigation, whereas tasks in the sandbox game domain include static object harvest tasks, dynamic target combat tasks, and highly long-horizon Creative Construction tasks.
> > > > - Regarding **operational frequency**, tasks in the household domain do not require high-frequency operations or control due to the static nature of objects. In contrast, many tasks in the sandbox game domain demand high-frequency controls to succeed, such as harvesting or combating animals.
> > > > - In terms of **object morphology**, objects in the household domain are more realistic, whereas those in the sandbox game domain exhibit a unique blocky style. General feature extractors trained on real-world data often fail to generalize directly, requiring domain-specific fine-tuning.
> > > >
> > > > Compared to the **web domain**, WebArena is a purely textual environment, which does not align well with our multi-modal retriever. Additionally, the VisualWebArena often requires detailed analysis of **lengthy HTML texts** to identify actionable icons, and the **state transitions** are significantly more dynamic.
> > > >
> > > > Considering the substantial gaps between different domains, directly demanding OOD generalization is **unrealistic**. A retriever model trained in the household domain is expected to **retain capabilities** for static objects that closely resemble real-world ones, while its critique ability for tasks involving extremely long and complex textual inputs or those with greater disparities is likely to be unreliable.
> > > >
> > > > In future work, we plan to evaluate the generalization capability within the same domain but under **unseen** environments and scenarios. For instance, we will train retrievers in the web domain, the open-ended sandbox game domain, and the household domain, respectively, to evaluate their generalization capability in novel environments and scenarios within each domain.

---

> ### Author Response · Authors · 2024-11-25
> **Response to the question 3.**
>
> **Q3.a.**  Could you clarify the meaning of AS (Average Steps) in your evaluation? Does it represent the number of iterations required for the agent to succeed once?
>
> **A3.a.** Thanks for your valuable question! AS (Average Steps) represents the **average number of steps** the agent takes to successfully complete a task. However, if the agent **fails** to complete the task, the task is marked as a **failure** after reaching the step limit $H$, and the AS value is **recorded as the step limit $H$**. This design has a notable **limitation**, as in practice, it is **not feasible** to allow the agent to operate **without a step limit**.
>
> **Q3.b.** Correlation Between AS and SR: Does Average Steps and Success Rate correlate? How do you interpret the results where AS shows minimal difference but SR differs more significantly?
>
> **A3.b.** In essence, AS (Average Steps) and SR (Success Rate) are correlated: the better the agent's performance, the higher the success rate and the fewer average steps needed to complete tasks. However, due to the aforementioned limitations in the step design, **AS fails to accurately reflect agent performance**. For instance, assume $H=100$. Method A, with **better** performance, completes the task in exactly 100 steps, while Method B, with **weaker** performance, fails to complete the task even with 200 steps. Although these methods show **significant differences** in success rates, their AS values are **identical**.
>
> In Table 4, the reason why AS shows minimal differences while SR varies significantly is related to this **limitation of average steps**. Tasks in AI2-THOR are more challenging than those in LEGENT, requiring more steps to complete.
>
> As a result, methods that **barely** complete tasks within the step limit appear **similar** to those that **fail** to complete tasks within the limit, resulting in **closely** clustered AS values but **significantly** larger differences in SR. This phenomenon is observed only in the AI2-THOR environment and not in LEGENT.
>
> **Q3.c.** Fairness of Using AS as a Metric: Considering the additional computational cost of incorporating an MLLM, would it be more appropriate to use the average inference time of the entire system instead of AS to ensure a fair comparison between methods?
>
> **A3.c.** Thank you for your valuable suggestion! We acknowledge that AS (Average Steps) is a **flawed** metric. In contrast, the average inference time per test provides a more **equitable** basis for comparison. However, it **shares the same limitation as AS**: when the agent reaches the upper limit $H$ without completing the task, the average inference time is also capped at this point.
>
> We have computed this metric, and the results are shown in the tables below (measured in seconds).
>
> Performance comparison of different methods：
>
> | Environment | PA      | LP      | SL      | RAP     | MART        |
> | ----------- | ------- | ------- | ------- | ------- | ----------- |
> | AI2-THOR    | 2404.48 | 2173.11 | 2223.01 | 2214.47 | **1857.10** |
> | LEGENT      | 355.72  | 377.10  | 315.56  | 310.59  | **208.42**  |
>
> Ablation studies of MART：
>
> | Environment | w/o Abstraction | Sim.+FTM | MART    |
> |-------------|------------------|----------|---------|
> | AI2-THOR    | 1963.48          | 1887.47  | **1857.10** |
> | LEGENT      | 278.75           | 284.07   | **208.42**  |
>
>
> **Q4.** Can You Provide Balanced Case Studies? To ensure a fair comparison, could you provide both the success and failure cases for each method (MLLM Retrieval and similarity retrieval) in your case studies? This would help understand each approach's strengths and weaknesses more comprehensively.
>
> **A4.** Thanks for your valuable suggestion to improve our case study! We have added balanced case study in **Appendix D**, encompassing both success and failure cases for each method, along with simplified reasoning processes for clarity.

---

> ### Comment · Reviewer_7byC · 2024-11-25
> **General opinion**
>
> While some issues may be challenging to address due to time constraints, I hope these can be explored and resolved in future work. In particular, I would appreciate a more **detailed discussion** in the main text about the areas I am concerned with:
>
> 1. Absence of a detailed ablation study and the potential need for more resources: Future work could benefit from including more detailed analyses on how different model components influence overall performance.
>
> 2. Fairness in method comparison: I recommend using a broader range of datasets and feature extraction methods in subsequent studies to demonstrate the advantages and limitations of the proposed method compared to traditional similarity retrieval approaches.
>
> 3. Concerns regarding hallucinations and stability: Given the uncertainty of model performance in unexpected scenarios, further validation of model stability when facing out-of-distribution data is advisable.
>
> 4. Importance of including diverse tasks: To enhance the robustness and applicability of the method, future work should consider incorporating tasks from different domains, such as webArena, textcraft, or Minecraft.
>
> Despite these points, I find the methodology presented in this paper to be innovative and solidly feasible within its field, representing a valuable contribution. I look forward to seeing these issues addressed.

---

> > ### Author Response · Authors · 2024-11-27
> >
> > We sincerely appreciate the reviewer taking the time to provide your insightful feedback. Would you consider raising your score if our responses have adequately addressed your comments?

---

> > > ### Comment · Reviewer_7byC · 2024-11-29
> > >
> > > While certain issues still need to be addressed in terms of system completeness and experimental thoroughness, I recognize the potential impact of resolving these issues on the method's applicability in more valuable domains. Despite these challenges, the innovation presented in the paper is commendable. Therefore, I will raise my score. I encourage continued effort and improvement in future work.

---

> > > > ### Author Response · Authors · 2024-11-29
> > > >
> > > > Dear Reviewer 7byC,
> > > >
> > > > Thank you for your quick response. Your constructive reviews have been incredibly helpful in enhancing our work!
> > > >
> > > > Thank you once again for your support and guidance.
> > > >
> > > > Best regards,
> > > >
> > > > Authors of Submission10453

---

> ### Author Response · Authors · 2024-11-27
>
> Thank you for your insightful response! Your suggestions are extremely helpful in enhancing the quality and reliability of our paper.
>
> > Modifying the original head of LLaVA raises concerns about the additional resources needed for labeling a novel dataset, especially if direct success rate measurements are unavailable to train your MLLM to maintain accurate classification or evaluation performance.
> >
>
> > Additionally, it would be beneficial to understand the specific datasets utilized, the labeling process, and whether each application requires training a new model on domain-specific datasets instead of leveraging SOTA LLMs like GPT-4 or even O1.
> >
>
> The first point to clarify is that **only** the retriever in our method requires training, while the downstream decision-making agent has no models that need training. Although several modules are designed, they all rely on a frozen MLLM and require no further training. The retriever, including its score head, is the only part of our method that requires training. The training data consists **exclusively** of preference pairs obtained through interaction with the environment, with **no other data involved**. Preference pairs are induced based on the success rate, which relies on a success signal. However, **success detection** is a fundamental interface of the simulator. Even without direct setup, it can be inferred through the simulator’s metadata using rule-based methods or MLLM. Implementing a success detector in the **real world** is not difficult. Several works have used VLMs for success detection. For instance, [1] in real-robot manipulation tasks, MiniGPT4 is fine-tuned to output a binary reward as a success detector. [2] Flamingo is fine-tuned on VQA data involving real-world tasks to determine task completion. Therefore, the effort required to construct the dataset is **acceptable**.
>
> > **The absence of a detailed ablation study** on planning LLM and MLM retrieval is notable. You cannot guarantee the **strict positive correlation** between the retrieval and planning processes, but the simple similarity method naturally has this feature. If the planning process is inaccurate, how do you ensure that the success rate used to label data for your retrieval model doesn’t compound errors, potentially reinforcing incorrect planning but correct retrieval scenarios?
> >
>
> Yes, our retriever is designed for the planning agent learning, because we believe that determining whether a trajectory is suitable depends on whether the planning can succeed. We use **5 trials** to eliminate the noise from downstream planning.
>
> As we have emphasized previously, similarity does not correlate with effectiveness. A better metric is the retrieved trajectory that results in a higher task success rate. If a trajectory seems relevant and offers useful information to the downstream agent but is difficult for the agent to comprehend—potentially due to its excessive length—leading to a low downstream task success rate, it cannot be considered a suitable trajectory.
>
> In future work, to evaluate the contribution of each component to overall performance, we plan to conduct a more comprehensive ablation study. This will include experiments with **other MLLM agents**, as well as analyzing various **components of the agent**, i.e. self-reflection mechanisms, prompt designs tailored to specific functions in action planning, and retrievers based on diverse base models, to assess their impact on overall performance.
>
> **Reference**
>
> [1] Yang, Jingyun, et al. "Robot fine-tuning made easy: Pre-training rewards and policies for autonomous real-world reinforcement learning." *2024 IEEE International Conference on Robotics and Automation (ICRA)*. IEEE, 2024.
>
> [2] Du, Yuqing, et al. "Vision-language models as success detectors." *arXiv preprint arXiv:2303.07280* (2023).

---

> ### Author Response · Authors · 2024-11-27
>
> We sincerely appreciate your insightful suggestion!
>
> Models such as CLIP and Sentence Transformer serve as feature extractors trained on large-scale datasets; however, their training primarily focuses on relations and does not account for success rates or effectiveness in decision-making tasks.
>
> A potential approach involves selecting high-win-rate trajectories from decision-making task scenarios, aligning them with the current task and observations to construct a domain-specific dataset, and then fine-tuning general feature extractors on these data.
>
> However, this method is both inefficient and significantly more expensive compared to ours. For a given task, sampling $K$ trajectories and interacting with MART can produce nearly $\binom{K}{2}$ sample pairs, whereas the alternative approach generates only one or a few high-win-rate items.
>
> In future work, we aim to further investigate this approach and conduct fair comparisons to highlight the advantages and limitations of our method in comparison to adapted traditional similarity retrieval approaches.

---

### Official Review · Reviewer_QSXR · 2024-10-29

**Soundness:** 3
**Presentation:** 3
**Contribution:** 2
**Rating:** 8
**Confidence:** 4

**Summary:**

This paper introduces MART, a novel trajectory retrieval paradigm leveraging interactive learning to enhance embodied agents’ task performance. MART utilizes interaction-based feedback to identify the most effective trajectories and constructs preference pairs based on trajectory comparisons. These preference pairs are used to fine-tune a Multimodal Large Language Model (MLLM) retriever, prioritizing trajectories that improve task outcomes. Additionally, MART introduces Trajectory Abstraction, a mechanism using MLLMs’ summarization capabilities to condense trajectory representations by reducing token counts while preserving essential information, enabling agents to better understand task-relevant details. Experimental results across diverse environments demonstrate that MART significantly improves success rates in unseen tasks compared to various baselines. This work bridges the gap between general-purpose MLLMs and embodied task requirements, offering a new paradigm for multimodal trajectory retrieval for embodied agents.

**Strengths:**

1. The problem addressed in this paper—decision-making in embodied tasks—is a crucial area.
2. The writing in this paper is clear and well-structured, with each paragraph presenting ideas in a logical, cohesive manner. The authors effectively communicate complex concepts, making the paper accessible and easy to follow.
3. This paper introduces an innovative multimodal retrieval tool and selects appropriate baselines to validate the effectiveness of the proposed method.

**Weaknesses:**

1. The methodology in this paper shows some similarities with previous works, such as LLM-Planner (arXiv:2212.04088 ) and P-RAG(	arXiv:2409.11279). While there are innovations in the retriever design, the overall novelty of the approach is somewhat reduced.
2. The paper lacks some detailed analysis and deeper insights. For instance, the ablation study does not include experiments explaining HOW Trajectory Abstraction contributes to performance improvement

**Questions:**

1. It would be better if the authors could further explain the specific innovations in the pipeline compared to prior works LLM-Planner and P-RAG, highlighting how the proposed approach diverges or improves upon these method.
2. Could the authors provide an explanation for HOW Trajectory Abstraction contributes to performance improvements?
3. If incorrect trajectories are retrieved, could this lead to catastrophic performance loss?

---

> ### Author Response · Authors · 2024-11-25
> **Thanks for your review! Here, we respond to your comments and address the issues. We hope to hear back from you if you have further questions!**
>
> **Q1.** Lacking innovation comparisons with prior works LLM-Planner and P-RAG.
>
> **A1.** Thanks for pointing out that! As noted in the appendix B of related works on embodied grounding, LLM-Planner and P-RAG fall under the category of LLM-as-Planner methods. These methods **pre-define a low-level controller**, where the **LLM or MLLM generates sub-goals**, and the controller translates these sub-goals into executable action sequences. In contrast, **our approach requires the MLLM to directly generate executable actions**.
>
> Regarding **retrieval** methods, both LLM-Planner and P-RAG rely on **similarity-based retrieval** methods, which **do not evaluate the effectiveness of trajectories for task completion**. Specifically, LLM-Planner uses a frozen BERT-base model to compute pairwise similarity and retrieve the top-K examples. And P-RAG computes the similarity of both the instruction and observation and sums their scores. In contrast, our method employs a multimodal large language model (MLLM) to predict the utility of trajectories for task completion, allowing the retrieval of the most effective ones.
>
> **Q2.** Providing an explanation for how Trajectory Abstraction contributes to performance improvements.
>
> **A2.** Trajectory Abstraction serves **two** critical purposes. **Firstly**, it compresses trajectories by **removing redundant observations and feedback** while **retaining key steps and information** pertinent to the current task. **Secondly**, it infers the **implicit rules of tasks** requiring multi-step operations from imperfect trajectories. These functionalities are demonstrated in both case studies presented in this work.
>
> In the **first case study**, Trajectory Abstraction effectively compressed a trajectory of 73 steps into only 5 significant milestones while preserving crucial information, such as the location of the target object. This abstraction significantly improved the agent's task performance. Under identical conditions across other modules, the MART framework achieved a success rate of 80% after five validations, compared to a 40% success rate achieved by the similarity-based method.
>
> In the **second case study**, involving a "heat" task, a 13-step imperfect trajectory obtained through exploration and trial-and-error was input into the Trajectory Abstraction module. The module effectively removed redundant actions and distilled the sequence into 6 essential steps. This refined trajectory enabled the agent to complete the task efficiently. Under identical conditions across other modules, the MART framework again achieved a success rate of 80% after five validations, whereas the similarity-based method failed entirely, with a success rate of 0%.
>
> These results underscore the critical advantages of Trajectory Abstraction in improving task efficiency and robustness in complex scenarios.
>
> Additionally, in the ablation studies, we compared the full MART approach with a version excluding the Trajectory Abstraction module (w/o Abstraction). The results demonstrate that Trajectory Abstraction notably improves the performance of embodied agents across environments.
>
> **Q3.** If incorrect trajectories are retrieved, could this lead to catastrophic performance loss?
>
> **A3.** Thanks for pointing out that!
>
> In the action planning module, we prompt the agent to analyze the provided reference trajectory result and evaluate its contribution to the current task. If the agent determines that the trajectory does not provide explicit assistance, it proceeds to complete the task based solely on the current task requirements and observations. In such cases, its success rate is expected to approximate that of a plain agent without reference trajectory input.
>
> To verify this, we conducted experiments where manually inputting trajectory that had low MART score and is manually verified to be irrelevant to the current task into the agent. The results demonstrated that the agent's success rate was comparable to that of the plain agent, confirming that irrelevant trajectories do not result in catastrophic performance loss.
>
> |                     | Irrelevant trajectory | Plain Agent |
> | ------------------- | --------------------- | ----------- |
> | SR $\uparrow$       | 0.21                  | 0.18        |
> | SR-Sub $\uparrow$   | 0.66                  | 0.63        |
> | AS $\downarrow$     | 149.30                | 159.66      |
> | AS-Sub $\downarrow$ | 41.75                 | 44.65       |

---

> > ### Comment · Reviewer_QSXR · 2024-11-28
> >
> > Thank you for your detailed responses to my comments and for the additional experiments and analyses. These efforts have helped clarify my concerns and provided me with a better understanding of your work. I will reconsider my score and adjust it accordingly based on this new information.

---

> > > ### Author Response · Authors · 2024-11-28
> > >
> > > Dear Reviewer QSXR,
> > >
> > > Thank you for your quick response. Your insightful reviews have been incredibly helpful in enhancing our work!
> > >
> > > Thank you once again for your support and guidance.
> > >
> > > Best regards,
> > >
> > > Authors of Submission10453

---

> ### Author Response · Authors · 2024-11-28
> **Looking Forward to the Feedback on the Rebuttal Response**
>
> Dear Reviewer QSXR,
>
> We sincerely appreciate your insightful feedback, which plays a crucial role in enhancing the quality of our work.
>
> As the rebuttal deadline approaches, we want to check whether our response addresses your concerns. If there are any additional questions or issues you would like us to address, please let us know. We would be grateful if you could provide stronger comments to support our work after reviewing our response.
>
> Thank you once again for your support and guidance.
>
> Best regards,
>
> Authors of Submission10453

---

### Official Review · Reviewer_EV8c · 2024-11-02

**Soundness:** 2
**Presentation:** 2
**Contribution:** 2
**Rating:** 6
**Confidence:** 3

**Summary:**

The paper presents a novel method, MLLM As ReTriever (MART), which aims to enhance the performance of embodied agents in complex environments by improving the retrieval of multimodal, task-relevant trajectory data. It uses expert trajectories as prompts for an MLLM agent, collects interactive feedback in the form of success rates, and organizes this data into preference pairs. These pairs are then used to fine-tune the MLLM retriever, enabling it to prioritize more effective trajectories for unseen tasks. The authors conducted experiments across two environments, demonstrating MART's success rates in unseen scenes.

**Strengths:**

1. This paper proposes a new retrieval-augmented MLLM agent, which finds the most matched expert reference trajectory to enhance current trajectory planning. To compress these multimodal trajectories with lots of images, actions, and feedback, Trajectory Abstraction (GPT-4o) is introduced to identify important trajectory milestones.

2. The experimental results demonstrate the effectiveness of MART in improving task success rates across different environments.

**Weaknesses:**

I am not an expert in this field, but I have several problems:

---

1. This work aims to tackle the challenge of grounding in environments. However, the literature review focuses more on embodied agents, memory retrieval in agents, and MLLM. These areas encompass a wide range of topics and, as a result, **overlook numerous specific and significant studies concerning how past embodied grounding approaches have tackled this issue**. In summary, I am unable to identify key references within this manuscript, and the innovation it presents remains obscure to me.

---
2. The proposed model is evaluated on AI2-THOR and LEGENT, with performance comparisons against Plain-Agent, LLaVA-Plain, Similarity+LLaVA, and RAP. However, it's worth noting that these comparative methods are predominantly variations of Multimodal Language Models (MLLM) or retrieval-based approaches, effectively serving as ablation studies. **This suggests that the absence of key comparisons with prior grounding techniques raises concerns regarding the efficacy of the proposed method's performance.**


---
3. What are the implementation details including model and training details?

**Questions:**

You can find all my inquiries detailed in the Weaknesses section.

---

> ### Author Response · Authors · 2024-11-25
> **Thanks for your review! Here, we respond to your comments and address the issues. We hope to hear back from you if you have further questions!**
>
> **Q1.** Overlooking numerous related works concerning embodied grounding approaches.
>
> **A1.** Thanks for your valuable suggestions!
>
> > This work aims to tackle the challenge of grounding in environments. However, the literature review focuses more on embodied agents, memory retrieval in agents, and MLLM.
>
> We would like to clarify that our work does not address a general grounding problem in environments; instead, it focuses on a more specific topic: **embodied grounding**. For clarification, an embodied agent can be broadly divided into two components: perception (**visual grounding**[1][2][3]) and decision-making (**embodied grounding**).
>
> In terms of perception, grounding in environments refers to visual grounding, which involves identifying the most relevant object or region in an image corresponding to a given language query. This aspect of grounding is typically addressed during the pre-training of MLLMs by **aligning visual and textual tokens**.
>
> In contrast, embodied grounding emphasizes adapting MLLMs to make decisions in downstream embodied environments or real-world scenarios. Specifically, embodied grounding seeks to understand **the effects of an agent’s actions** on environmental dynamics and **how to generate action sequences** to achieve a given task. Unlike visual grounding, embodied grounding capabilities cannot be acquired through pre-training due to a significant gap between pre-training datasets and specific execution scenarios, such as safety constraints and object affordances.
>
> Our work focuses on the field of embodied grounding, leveraging a retrieval-augmented generation framework. Therefore, the related work section primarily emphasizes embodied agents and memory retrieval, aligning with the specific focus of our research.
>
> > Overlook numerous specific and significant studies concerning how past embodied grounding approaches have tackled this issue.
>
> We sincerely thank the reviewer for pointing out the missing aspects in our related work section.  In response, we newly added related works on embodied grounding in Appendix B. Approaches to addressing embodied grounding can be broadly categorized into the following types:
>
> **RL**: Reinforcement learning (RL) trains an agent’s policy through interaction with the environment, making the agent inherently grounded in the environment[4][5]. However, RL typically requires extensive interaction with environments and often suffers from instability, making it unsuitable for MLLMs.
>
> **VLA**: These methods focus on fine-tuning vision-language models (VLMs) using expert datasets collected from embodied environments, such as PaLM-E[6] and RT-2[7]. These methods demand a significant amount of high-quality trajectory data for training.
>
> **LLM-as-Planner**: These methods leverage Large Language Models (LLMs) or Multimodal Large Language Models (MLLMs) to generate high-level plans, which are then translated into executable action sequences by low-level controllers[8][9]. A key limitation of these methods is their reliance on a predefined skill library, which restricts the scope of the agent’s capabilities. Besides, acquiring a skill library might require additional RL or IL training or prior knowledge about the environment[10].
>
> **Retrieval-Augmented Agent**: This category involves integrating task trajectory data into the prompts provided to MLLMs[11]. These trajectory data, rich in grounding information about the environment, enable agents to perform tasks effectively. Retrieval-augmented methods usually demonstrate greater sample efficiency compared to RL and VLA, thanks to the use of an explicit memory buffer. Our work falls into this category.

---

> ### Author Response · Authors · 2024-11-25
> **Response to the remaining two questions.**
>
> **Q2.** Lacking key comparisons with prior embodied grounding techniques.
>
> **A2.** Thanks for your valuable suggestions to expand our baselines!
>
> > **This suggests that the absence of key comparisons with prior grounding techniques raises concerns regarding the efficacy of the proposed method's performance.**
>
> We appreciate the reviewer for highlighting the comparison with prior grounding techniques. As explained earlier, we focus on addressing the **embodied grounding of MLLMs**, so we newly added previous embodied grounding techniques baselines for comparison, including **LLM-as-planner and VLA**. However,  LLM-as-planner approaches generally require a **predefined skill library** which is not available in the two embodied environments, so we do not include them in our newly added comparison.
>
> To provide a more comprehensive comparison, we have included the VLA method as a baseline in our evaluation.
>
> The LEGENT paper conducted experiments on the LEGENT environment using the VLA method. VILA-7B [12] was adopted as the backbone model for its ability to handle interleaved inputs. The model was trained on **thousands of high-quality trajectory** datasets to predict the current action based on task descriptions and the interleaved context of prior observations and actions. The model was evaluated on 100 trajectories spanning all four tasks. The experimental results, compared to our proposed method, are summarized in the table below:
>
> |                 | Success Rate |
> | --------------- | ------------ |
> | VILA-7B-Sep 1K  | 0.41         |
> | VILA-7B-Sep 10K | 0.78         |
> | VILA-7B-Joint   | 0.81         |
> | MART            | **0.87**     |
>
> VILA-Sep denotes models fine-tuned separately for each task, whereas VILA-Joint refers to models trained jointly on all tasks. The paper does not specify the amount of data used to train VILA-Joint, but it is likely comparable to or greater than that used for VILA-Sep. In comparison, our method employs just 40 trajectories for training and 32 for testing.
>
> **Q3.** Providing implementation details including model and training details.
>
> **A3.** Thanks for pointing out that! We complemented the implementation details about the model, training process and implementation pipeline for the whole framework in **Appendix A**. We commit that all code, including the model, training process, benchmark tasks, and simulator, will be released upon acceptance.
>
> **Reference**
>
> [1] Lai, Xin, et al. "Lisa: Reasoning segmentation via large language model." *Proceedings of the IEEE/CVF Conference on Computer Vision and Pattern Recognition*. 2024.
>
> [2] Kazemzadeh, Sahar, et al. "Referitgame: Referring to objects in photographs of natural scenes." *Proceedings of the 2014 conference on empirical methods in natural language processing (EMNLP)*. 2014.
>
> [3] Nagaraja, Varun K., Vlad I. Morariu, and Larry S. Davis. "Modeling context between objects for referring expression understanding." *Computer Vision–ECCV 2016: 14th European Conference, Amsterdam, The Netherlands, October 11–14, 2016, Proceedings, Part IV 14*. Springer International Publishing, 2016.
>
> [4] Schulman, John, et al. "Proximal policy optimization algorithms." *arXiv preprint arXiv:1707.06347* (2017).
>
> [5] Haarnoja, Tuomas, et al. "Soft actor-critic: Off-policy maximum entropy deep reinforcement learning with a stochastic actor." *International conference on machine learning*. PMLR, 2018.
>
> [6] Driess, Danny, et al. "Palm-e: An embodied multimodal language model." *arXiv preprint arXiv:2303.03378* (2023).
>
> [7] Brohan, Anthony, et al. "Rt-2: Vision-language-action models transfer web knowledge to robotic control." *arXiv preprint arXiv:2307.15818* (2023). [8] Song, Chan Hee, et al. "Llm-planner: Few-shot grounded planning for embodied agents with large language models." *Proceedings of the IEEE/CVF International Conference on Computer Vision*. 2023.
>
> [9] Xu, Weiye, et al. "P-RAG: Progressive Retrieval Augmented Generation For Planning on Embodied Everyday Task." *Proceedings of the 32nd ACM International Conference on Multimedia*. 2024.
>
> [10] Lifshitz, Shalev, et al. "Steve-1: A generative model for text-to-behavior in minecraft." *Advances in Neural Information Processing Systems* 36 (2024).
>
> [11] Kagaya, Tomoyuki, et al. "Rap: Retrieval-augmented planning with contextual memory for multimodal llm agents." *arXiv preprint arXiv:2402.03610* (2024).
>
> [12] Lin, Ji, et al. "Vila: On pre-training for visual language models." Proceedings of the IEEE/CVF Conference on Computer Vision and Pattern Recognition. 2024.

---

> ### Author Response · Authors · 2024-11-28
> **Looking Forward to the Feedback on the Rebuttal Response**
>
> Dear Reviewer EV8c,
>
> We sincerely appreciate your constructive feedback, which plays a crucial role in enhancing the quality of our work.
>
> As the rebuttal deadline approaches, we want to check whether our response addresses your concerns. If there are any additional questions or issues you would like us to address, please let us know. We would be grateful if you could provide stronger comments to support our work after reviewing our response.
>
> Thank you once again for your support and guidance.
>
> Best regards,
>
> Authors of Submission10453

---

> ### Author Response · Authors · 2024-12-03
> **Looking Forward to the Feedback on the Rebuttal Response**
>
> Dear Reviewer EV8c,
>
> We sincerely appreciate your constructive feedback, which plays a crucial role in enhancing the quality of our work.
>
> This is a friendly reminder that only **half a day** remains in the rebuttal period, after which authors and reviewers will no longer be able to communicate. We want to check whether our response addresses your concerns. If there are any additional questions or issues you would like us to address, please let us know. We would be grateful if you could provide stronger comments to support our work after reviewing our response.
>
> Thank you once again for your support and guidance.
>
> Best regards,
>
> Authors of Submission10453

---

### Official Review · Reviewer_CWhu · 2024-11-03

**Soundness:** 3
**Presentation:** 3
**Contribution:** 3
**Rating:** 6
**Confidence:** 3

**Summary:**

The paper introduces a novel method, MLLM As ReTriever (MART), which enhances the performance of embodied agents by utilizing interaction data to fine-tune a multimodal language model (MLLM) retriever. The method aims to improve the effectiveness of retrieved trajectories for specific tasks, and it introduces Trajectory Abstraction to summarize trajectories while preserving key information. The experimental results show significant improvements in task success rates in unseen scenes compared to baseline methods.

**Strengths:**

1. The paper proposes a unique method for enhancing multimodal retrieval in embodied agents by leveraging preference learning and trajectory abstraction. This approach addresses a critical gap in current retrieval methods that often focus on surface-level similarities.
2.  The introduction of Trajectory Abstraction is a valuable contribution. It effectively reduces the complexity of trajectories while maintaining essential information
3. The experimental results are comprehensive and demonstrate the effectiveness of the proposed method across various environments. The improvements in task success rates in unseen scenes are particularly noteworthy.

**Weaknesses:**

1. While the paper demonstrates the effectiveness of MART in several environments, the scope of experiments could be expanded to include more diverse and challenging scenarios to further validate the robustness of the method.
2. The paper does not provide a detailed analysis of the computational complexity and resource requirements of the proposed method. This information is crucial for practical implementation and comparison with existing methods.
3. The paper mentions the use of interaction data but does not delve into the specifics of how user interactions are collected and processed. More details on this aspect would enhance the clarity and reproducibility of the method.
4. Provide more detailed implementation guidelines and discuss potential limitations and solutions. This would make the method more accessible to other researchers and practitioners.

**Questions:**

See above.

---

> ### Author Response · Authors · 2024-11-25
> **Thanks for your review! Here, we respond to your comments and address the issues. We hope to hear back from you if you have further questions!**
>
> **Q1.** While the paper demonstrates the effectiveness of MART in several environments, the scope of experiments could be expanded to include more diverse and challenging scenarios to further validate the robustness of the method.
>
> **A1.** Thanks for your valuable suggestions to expand our experiment scenarios! We validated our approach on the **more diverse and challenging ReALFRED [1]** environment, where tasks require cross-room navigation and the environment is more realistic. The result of experiments on ReALFRED is shown below, and details are in Appendix C.
>
> |       | PA    | LP    | SL    | RAP   | MART  |
> |-------|-------|-------|-------|-------|-------|
> | SR ↑  | 0.25  | 0.20  | 0.26  | 0.27  | **0.37**  |
> | SR-Sub ↑ | 0.50  | 0.44  | 0.52  | 0.53  | **0.58**  |
> | AS ↓  | 101.70| 87.93 | 89.79 | 87.89 | **78.48** |
> | AS-Sub ↓ | 28.43 | 24.32 | 24.84 | 24.31 | **21.71** |
>
> **Q2.** Lacking detailed analysis of computational complexity and resource requirements.
>
> **A2.** Thanks for pointing this out! In more detail, for the **training** phase, we use 8 A100s (40GB) to train the retriever model with **Lora** and only train **one epoch** to prevent overfitting, and since we have replaced the language model head into a Bradley-Terry score head, our retriever model **generates only one new token** at a time, which greatly reduce the computation and time cost of both train and evaluation phrases. For the **evaluation** phase, we just used a computer with **a single 4090 GPU** to experiment, and for each query, we input **multiple memory trajectories as a batch** to speed up, so that the complexity is $O(n/b)$, where $n$ means the number of memory trajectories and $b$ means the number of trajectories in batch.
> We also compared our inference time with the similarity-based method under the same conditions, and the results are summarized in the table.
>
> |                        | Similarity-base Retrieval | MLLM retriever | Action Planning |
> |------------------------|---------------------------|----------------|-----------------|
> | Seconds per query      | 0.05                      | 0.43           | 15.06           |
> | Repetitions            | 1                         | 1              | multiple         |
>
> However, for the entire pipeline, the main **bottleneck** is the action planning part. That is, for each task, we only perform **one retrieval at the beginning** and then perform **multiple action planning** processes to complete the task. The average time spent on one action planning is **15.06** seconds, and this time has to **repeat dozens of times**. Although the retrieval stage of  MART has a higher time complexity compared to the similarity-based retrieval method, the actual difference is minimal, as the primary time bottleneck lies in action planning.
>
> **Q3.** Lacking collecting and processing details of user interactions.
>
> **A3.** Thank you for your question. There seems to be a misunderstanding. In the article, "interaction" refers to the agent interacting with the environment to obtain the winning rate as feedback, rather than user interaction. However, we appreciate your suggestion, as it provides valuable direction for future work. We will explore how MART can be integrated into a human-in-the-loop setting.
>
> **Q4.1.** Providing more detailed implementation guidelines.
>
> **A4.1.** Thank you for pointing this out! We complemented the implementation details about the model, training process, and implementation pipeline for the whole framework in **Appendix A**. We guarantee that **all code will be released** upon acceptance.
>
> **Q4.2.** Discussing potential limitations and solutions.
>
> **A4.2.** Thank you for pointing this out! One limitation **already discussed in our conclusion** is the MLLM's limited context window, restricting input to a single trajectory at a time, similar to one-shot learning. Future work could explore few-shot learning with multiple trajectories, potentially enabling skill splicing and improved performance on complex, longer-horizon tasks. Another potential limitation is the usage of closed-source MLLM (GPT-4o) in our agent framework. Each module in our agent design requires GPT-4o to process a large number of images and text tokens, which requires a long time to wait. We are considering replacing it with an open-source MLLM such as qwen2-vl and deploying it locally, which will reduce the waiting time for communication and make our method available in all regions.
>
> **Reference**
>
> [1] Kim, Taewoong, et al. "ReALFRED: An Embodied Instruction Following Benchmark in Photo-Realistic Environments." European Conference on Computer Vision. Springer, Cham, 2025.

---

> ### Author Response · Authors · 2024-11-28
> **Looking Forward to the Feedback on the Rebuttal Response**
>
> Dear Reviewer CWhu,
>
> We sincerely appreciate your invaluable feedback, which plays a crucial role in enhancing the quality of our work.
>
> As the rebuttal deadline approaches, we want to check whether our response addresses your concerns. If there are any additional questions or issues you would like us to address, please let us know. We would be grateful if you could provide stronger comments to support our work after reviewing our response.
>
> Thank you once again for your support and guidance.
>
> Best regards,
>
> Authors of Submission10453

---

> ### Author Response · Authors · 2024-12-03
> **Looking Forward to the Feedback on the Rebuttal Response**
>
> Dear Reviewer CWhu,
>
> We sincerely appreciate your invaluable feedback, which plays a crucial role in enhancing the quality of our work.
>
> This is a friendly reminder that only **half a day** remains in the rebuttal period, after which authors and reviewers will no longer be able to communicate. We want to check whether our response addresses your concerns. If there are any additional questions or issues you would like us to address, please let us know. We would be grateful if you could provide stronger comments to support our work after reviewing our response.
>
> Thank you once again for your support and guidance.
>
> Best regards,
>
> Authors of Submission10453

---

### Meta-Review · Area_Chair_4yhC · 2024-12-20

**Metareview:**

**Summary**

The paper proposes to use multi-modal large language models (MLLM) to retrieve trajectories for embodied tasks by fine-tuning the MLLM to judge the effectiveness of the retrieved trajectory.  To capture important information in the trajectory, the paper introduces a mechanism (using MLLMs) to create a trajectory abstraction (e.g. summarization) that extract key milestones from the complete trajectory and augment it with additional information such as a description, feedback, and actions taken between the current milestone and the previous milestone.  Experiments are conducted on two environments (AI2-THOR, LEGENT) to compare the effectiveness of the proposed approach to alternatives.

The main contributions of the work are the proposed model (MART) that fine-tunes the MLLM, the notion of the trajectory abstraction, and the experiments evaluating the effectiveness of the proposed approach.

**Strengths**

Reviewers noted the following strengths of the work:
1. Decision making in embodied area is important [QSXR]
2. Proposed model of retrieval-augmented MLLM agent and the idea of trajectory abstraction seems to be useful and effective [CWhu,EV8c,QSXR,7byC]
3. Sufficient experiments demonstrating the effectiveness of the proposed approach [CWhu,EV8c,QSXR,7byC]
4. Clear and well-structured paper [QSXR]

**Weaknesses**

During the review process, the reviewers noted the following weaknesses.
1. Scope of experiments is relatively limited
   - More challenging scenarios [CWhu]
   - Additional baselines [EV8c]
2. Lack of information on computational and resource costs [CWhu,7byC]
3. Limited information on how user interactions are collected and processed [CWhu]
4. Additional details about implementation [CWhu, EV8c], evaluation [7byC] and discussion of limitations [CWhu,7byC]
5. Limited discussion of how prior owrk tackled the embodied grounding problem [EV8c] and clarifications of innovations [QSXR]

Many of these where addressed by the authors during the rebuttal period, causing the reviewers to increase their scores.


**Recommendation**

The reviewers are overall positive on the work, and their concerns were addressed during the author response period.  The AC agrees with the assessment of the reviewers that the work is interesting and well-presented, and recommends acceptance.

**Additional Comments On Reviewer Discussion:**

Reviewers are overall positive on this work.  The work initially received one 5 and three 6's.  Based on reviewer questions and concerns, the authors provided additional experiments, missing details, and improved / added discussion of related work and limitations.  Reviewers indicated their concerns were addressed and after the author response period, reviewers increased their scores to three 6's and one 8.

---

### Decision · Program_Chairs · 2025-01-22

Accept (Poster)